# Online Continual Learning with Dynamic Label Hierarchies

Xinrui Wang [1 2 3]  Bartłomiej Twardowski [3 4 5]  Alexandra Gomez-Villa [3 4]  Shao-Yuan Li [1 2 6 7]  Songcan Chen [1 2]

## Abstract

Online Continual Learning (OCL) aims to learn from endless non-stationary data streams, yet most existing methods assume a flat label space and overlook the hierarchical organization of real-world concepts that evolves both horizontally (sibling classes) and vertically (coarse or fine categories). To better reflect this context, we introduce a new problem setting, DHOCL (Online Continual Learning from Dynamic Hierarchies), where taxonomies evolve across granularities and each sample provides supervision at a single hierarchical level. In this setting, we find two fundamental issues: (i) partial supervision under mixed granularities provides only point-wise signals over an evolving path-wise hierarchy, which constrains plasticity and undermines cross-level semantic consistency, and (ii) the dynamically evolving hierarchies induce granularity-dependent interference, destabilizing popular replay and regularization mechanisms and thereby exacerbating catastrophic forgetting. To tackle these issues, we propose HALO (Hierarchical Adaptive Learning with Organized Prototypes), which adaptively combines complementary classification heads, regularized by organized learnable hierarchical prototypes, enabling rapid adaptation, hierarchical consistency, and structured knowledge consolidation as the taxonomy evolves. Extensive experiments on multiple benchmarks demonstrate that HALO consistently outperforms existing methods across hierarchical accuracy, mistake severity, and continual performance.

[1] College of Computer Science and Technology, Nanjing University of Aeronautics and Astronautics [2] MIIT Key Laboratory of Pattern Analysis and Machine Intelligence [3] Computer Vision Center, Spain [4] Computer Sciences Department, Universitat Autonoma de Barcelona, Spain [5] IDEAS Research Institute, Warsaw, Poland [6] State Key Laboratory for Novel Software Technology, Nanjing University [7] Joint Laboratory of Spatial intelligent Perception and Large Model Application, China. Correspondence to: Songcan Chen <s.chen@nuaa.edu.cn>.

*Proceedings of the 43rd International Conference on Machine Learning*, Seoul, South Korea. PMLR 306, 2026. Copyright 2026 by the author(s).

## 1. Introduction

Online Continual Learning (OCL) has emerged as a critical paradigm for deploying AI systems in dynamic environments where data arrives in a non-stationary stream (Gunasekara et al., 2023; Zhou et al., 2024). However, existing OCL methods generally assume a "flat" label space where all categories are treated as independent and semantically equidistant, which obviously clashes with how knowledge is organized in real world. Online biodiversity monitoring makes this issue concrete. For platforms like iNaturalist and Global Biodiversity Information Facility (GBIF), their taxonomic frameworks [1] are inherently fluid. As new species are discovered and phylogenetic relationships are revised, the taxonomy must track these shifts over time. Meanwhile, real-world observations often exhibit diverse granularity: depending on an observer's expertise, from coarse level like *Bird* to species identifiers like *Monarcha melanopsis*.

These realities call for an OCL formulation capable of handling hierarchical structures and dynamically evolving label spaces. To this end, we propose Dynamic Hierarchical Online Continual Learning (DHOCL), which transcends the conventional "flat" label space assumption and, crucially, abandons the rigid, predefined curricula (e.g., strict coarse-to-fine progression)in prior studies (Abdelsalam et al., 2021; Lee et al., 2023). As illustrated in Figure 1, DHOCL allows data instances to arrive with annotations at arbitrary hierarchical levels at any time, reflecting the diverse and uncertain nature of real-world supervision. For example, a model might first encounter a specific breed like *Siamese Cat*, then a broad, unrelated category like *Plant*, and later an intermediate ancestor *Feline* of the cat. This paradigm jointly captures horizontal expansion (the arrival of novel semantic categories) and vertical expansion (the refinement or abstraction of existing branches), effectively simulating the natural evolution of taxonomies. By bridging these gaps, DHOCL emerges as a uniquely flexible and realistic setting within the current CL landscape.

Despite its realism, DHOCL imposes significant challenges. Besides the structural overhead of maintaining an evolving taxonomy, we identify two fundamental hurdles beyond : (1) *Partial Supervision and Structural Decoupling.* In DHOCL, each instance provides only a "point-wise" glimpse (a single-

[1] https://hosted-datasets.gbif.org

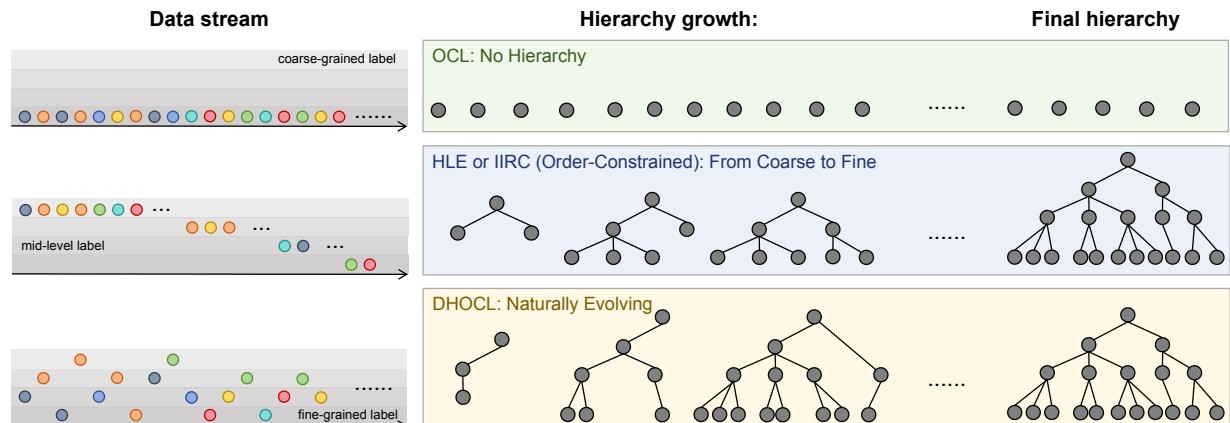

*Figure 1.* Hierarchy formation in three existing settings (colors denote classes and ellipses indicate time continuation). **Top:** OCL (Koh et al., 2022) uses flat labels only, yielding a flat final label space. **Middle:** HLE and IIRC (Lee et al., 2023; Abdelsalam et al., 2021) operate under a strict coarse-to-fine curriculum, in which parent classes must be introduced before their descendants, yielding a predefined and fixed hierarchy (typically balanced). **Bottom:** DHOCL allows labels at arbitrary hierarchical levels to arrive at any time, without assuming a predefined curriculum or fixed taxonomy. As a result, the hierarchy emerges and evolves dynamically, leading to an organically grown, potentially unbalanced structure that cannot be represented under the assumptions of prior settings.

level label) of what is inherently a "path-wise" structure (the full taxonomic lineage). When this partial supervision is coupled with the dynamic evolution of the hierarchy, it prevents the propagation of semantic constraints across granularities. This hinders the model from maintaining cross-level consistency, ultimately leading to a fragmented latent space where taxonomic branches become structurally decoupled. (2) *Granularity-Induced Interference.* Knowledge retention is intrinsically more complex under this mixed granularity. We observe that the model's performance and stability are not uniform across the hierarchy; rather, it is driven by the disparate learning and forgetting rates inherent to different taxonomic depths. This temporal asynchrony triggers semantic interference, where the learning and consolidation of coarse-grained details sometimes destabilizes fine-grained nuances, and vice versa. Existing replay and distillation strategies, designed for flat spaces, fail to account for this hierarchical cross-talk, leading to biased regularization.

To address such multifaceted challenges, we propose **HALO** (Hierarchical Adaptive Learning with Organized Prototypes), a novel framework for DHOCL. At its core, HALO employs learnable hierarchical prototypes that serve as the "topological glue" of the latent space. By explicitly constraining the inter-level representation geometry to mirror the taxonomic tree, these prototypes encourage that updates from any single level propagate structural constraints throughout the lineage, preventing decoupling. Meanwhile, these prototypes can also function as a structured and granular repository of historical knowledge, providing a stable anchor for representation consolidation as the hierarchy evolves over time. Building upon this stabilized feature space, we further design a hierarchical adaptive ensemble module (PredLA), a classifier-level strategy that harmonizes

asynchronous learning and forgetting dynamics. PredLA balances rapid adaptation with long-term stability, effectively mitigating inter-level interference and preserving fine-grained discriminability across all semantic levels.

In summary, our work's contributions are threefold. [2]

- We formalize DHOCL as a more realistic and general setting that captures the dynamic evolution of hierarchical knowledge in streaming data, filling a critical gap between existing hierarchical classification and OCL.

- We develop the framework HALO with hierarchical adaptive aggregation and prototype-based regularization, which uniquely addresses the challenges brought by dynamic label hierarchies in DHOCL.

- We establish a comprehensive evaluation protocol with multiple metrics to properly assess hierarchical continual learning performance, and validate our approach across diverse benchmarks (CIFAR-100, Aircraft, CUB-200, iNaturalist, ImageNet-H) that cover various hierarchical structures and domains.

## 2. Problem Setting and Preliminaries

**Beyond Coarse-to-Fine Curricula.** Prior hierarchical CL works (e.g., HLE (Lee et al., 2023), IIRC (Abdelsalam et al., 2021)) typically assume a strict deductive (coarse-to-fine) curriculum. We argue that this rigid assumption is often violated in practice due to three factors: (1) *Expertise Diversity:* Annotation granularity is often driven

---

[2]Our code is available at https://github.com/wxr99/HALO_ICML26

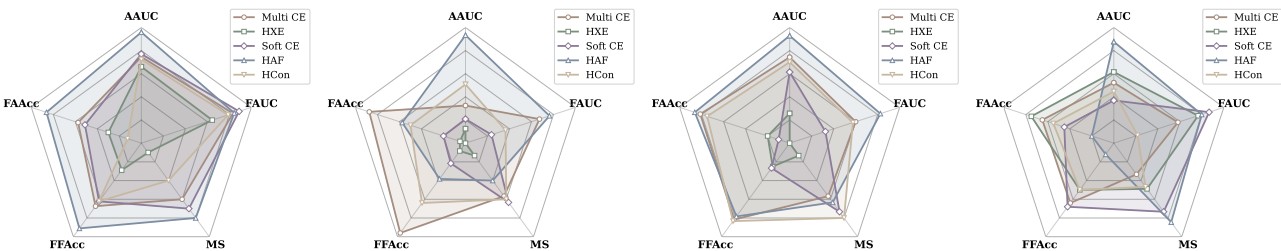

*Figure 2.* Performance comparison under DHOCL on CIFAR-100 (Krizhevsky et al., 2009), FGVC-Aircraft (Maji et al., 2013), CUB-200 (Wah et al., 2011), and iNaturalist (Van Horn et al., 2018) (from left to right). All methods employ reservoir sampling-based memory buffers with sizes of 1000, 1000, 1000, and 5000, respectively. Evaluation is across five metrics capturing overall performance, fine-grained accuracy, and mistake severity.

by the observer's expertise (e.g., a layperson's "Bird" vs. an expert's "Monarcha") rather than a fixed schedule. (2) *Inductive Growth:* Knowledge expansion is frequently inductive, where specific species are discovered long before their taxonomic families are established. (3) *Structural Openness:* Unlike "closed-world" taxonomies that require a full blueprint upfront, DHOCL accommodates "out-of-vocabulary" branches that emerge organically. In this light, HLE and IIRC are degenerate cases of DHOCL where the stream is artificially restricted to a top-down temporal order.

**Stream Definition.** This work considers a task-free continual learning setting where the model receives the data stream $\mathcal{D} = \{(x_1, y_1), (x_2, y_2), \dots\}$ in which $x_j \in \mathcal{X}$ is the input and $y_j = (c_j, h_j)$ represents a class label $c_j \in \mathcal{Y}$ at granularity level $h_j$. To simulate realistic yet reproducible problem settings, we partition the dataset by fine-grained classes into groups (as in class-incremental learning) and then blur boundaries between consecutive groups via category overlap, yielding a stream with no sharp task transitions. Meanwhile, each sample is annotated at a single granularity $h_j$ (e.g., "Dog"), while labels at other levels (e.g., the super-class "Animal" or the fine-grained breed "Golden Retriever") are unspecified, reflecting the practical scarcity of exhaustive or uniformly granular annotations.

**Dynamic Hierarchy Construction.** Unlike prior works that rely on a predefined taxonomy, DHOCL constructs the label tree $\mathcal{T}_t = (\mathcal{V}_t, \mathcal{E}_t)$ on-the-fly. We assume the global hierarchy is hidden; the model must resolve semantic relations as new classes emerge. While such relations could theoretically be induced from raw data (an orthogonal challenge), we adopt a more pragmatic incremental knowledge retrieval process. Specifically, when a new class $c_j$ arrives at time $t$, the model attempts to link it to the existing vocabulary $\mathcal{V}_{t-1}$ by querying an external knowledge source $\mathcal{K}$ (e.g., WordNet (Fellbaum, 1998), Expert System or large language models like GPT (Achiam et al., 2023)):

$$\mathcal{R}(c_j) = \mathcal{K}(c_j, \mathcal{V}_{t-1}),$$

where $\mathcal{R}(c_j)$ returns the set of local parent-child links used

to update the edge set $\mathcal{E}_t$. This construction process is strictly governed by two realistic constraints: (1) *Structural Incompleteness*, which dictates that the model never accesses a global blueprint. At any time $t$, it only maintains a partial sub-tree $\mathcal{T}_t$ spanning the observed classes, rendering the hierarchy a growing graph where nodes and edges are added or refined dynamically rather than following a fixed template. (2) *Temporal Asynchrony*, which recognizes that the semantic relations $\mathcal{R}(c_j)$ may not be immediately resolvable upon the arrival of $c_j$. This necessitates the model's ability to process "unanchored" classes that temporarily exist as isolated nodes outside the tree structure, better reflecting the incremental and often fragmented nature of knowledge acquisition in open-world environments.

As the label tree $\mathcal{T}_t$ evolves, the model needs to maintain a specific classifier for each observed granularity level. Let $f_\theta : \mathcal{X} \to \mathbb{R}^d$ be the backbone feature extractor. For each level $h \in \mathcal{H}_t$, a head $g_\phi^h : \mathbb{R}^d \to \mathbb{R}^{|\mathcal{V}_t^h|}$ produces logits over classes $\mathcal{V}_t^h \subseteq \mathcal{V}_t$ at that level, yielding:

$$\mathbf{p}^h(x) = \text{softmax}\big(g_\phi^h(f_\theta(x))\big) \in \Delta^{|\mathcal{V}_t^h|}. \tag{1}$$

Here, $\Delta^{|\mathcal{V}_t^h|}$ denotes the probability simplex. As new labels arrive, both the set of levels and the class sets per level expand, and the corresponding heads are updated.

**Evaluation protocol.** DHOCL targets accurate and consistent classification over all encountered classes at all granularity levels throughout training, and we evaluate accordingly. At regular intervals (e.g., every $N$ samples), we test on a held-out set covering all seen classes and record accuracy averaged across hierarchy levels, fine-grained accuracy, and mistake severity. Mistake severity follows (Bertinetto et al., 2020) which penalizes errors by semantic distance in the hierarchy. We then summarize these trajectories over time by reporting *AAUC*, *FAUC*, and *MS*, following standard practice in OCL (Koh et al., 2022; Moon et al., 2023), where *AAUC* is the area under the time curve of accuracy averaged across hierarchy levels, *FAUC* is the area under the time curve of fine-grained accuracy, and *MS* is the area under the

mistake-severity curve. We additionally report *FFAcc* and *FAAcc*, which denote the final accuracies at the fine-grained level and averaged over all hierarchy levels, respectively. Collectively, these five metrics reflect both plasticity and stability, span performance from coarse to fine granularity, and measure the semantic severity of errors.

## 3. OCL with Dynamic Hierarchies

While OCL and Hierarchical Classification are well-studied individually, their intersection in DHOCL introduces unique challenges (For a systematic review for these two topics, please refer to Appendix A). In this section, we analyze how and why existing paradigms struggle under the dual pressure of *streaming data* and *evolving taxonomies*. We first focus on loss-based hierarchical-aware approaches that naturally support DHOCL. We exclude embedding-based methods and architecture-specialized methods that requires continuous restructuring of embedding spaces or model components under evolving taxonomies – operations that are difficult to integrate directly into DHOCL. Concretely, our evaluation includes: *Multi-CE* (Wu et al., 2016), which applies level-wise cross-entropy (CE) losses; *HXE* (Bertinetto et al., 2020), which reweights losses via conditional hierarchical probabilities; *Soft-CE* (Bertinetto et al., 2020), which softens labels based on hierarchical distance; *HAF* (Garg et al., 2022), which enforces cross-level consistency; and *HCon* (Zhang et al., 2022), which introduces contrastive learning for structured embeddings. All methods are evaluated on CIFAR-100, FGVC-Aircraft, CUB-200, and iNaturalist, partitioned into ten incremental groups following the DHOCL setup described in Section 2.

As shown in Figure 2, no loss dominates across all metrics. Specifically, these hierarchical-aware losses fluctuate between marginal gains and performance degradation compared with the naive CE loss. We argue that this stems from an implicit conflict between current and historical optimization objectives within an evolving taxonomy. For instance, two classes initially defined as siblings may have their semantic distance drastically increased by the sudden emergence of a new intermediate parent node, leading to a phenomenon we term "supervisory jitter", manifested as abrupt shifts in the losses. This instability is further amplified by two key factors: First, the *inherent rigidity of output-end regularizations*. Even when softened, constraints based on logits or probability distributions lack the necessary flexibility confronting such topological updates. Second, the *volatility of structural transitions* which is exacerbated by the streaming nature of the task. Unlike standard continual learning, the more frequent arrival of new classes in a data stream triggers intense and recursive reorganization of the taxonomy. Enforcing strict hierarchical supervision during such volatile periods becomes counter-productive,

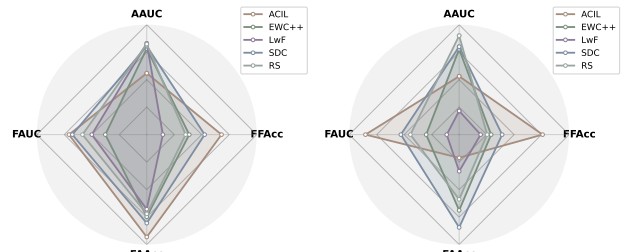

*Figure 3.* Performance comparison of four anti-forgetting strategies under DHOCL on CIFAR-100 (left) and iNaturalist (right).

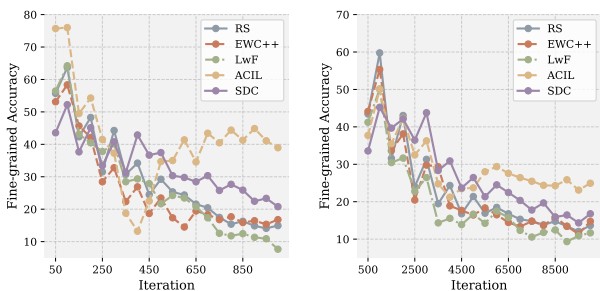

*Figure 4.* Learning dynamic of different methods on CIFAR-100 (left) and iNaturalist (right) under DHOCL. Performance is evaluated on fine-grained Acc, which reflects forgetting resistance.

as it forces the model to quickly converge toward "moving targets" within an unstable structural landscape.

Having established the limitations of hierarchical losses, we now examine whether standard anti-forgetting strategies can compensate for these deficiencies. To this end, we evaluate four representative baselines: parameter regularization based EWC++ (Chaudhry et al., 2018), logit distillation based LwF (Li & Hoiem, 2017), classifier based ACIL (Zhuang et al., 2022), prototype based SDC (Yu et al., 2020), and a naive reservoir sampling experience replay baseline denoted as RS (Rolnick et al., 2019) (Please refer to Sec A for more details of related works). All the four anti-forgetting baselines are integrated into a RS memory-based online setup. As shown in Fig. 3 and Fig. 4, none of these methods is able to maintain a consistent balance across hierarchical metrics under an evolving taxonomy. We argue that this sub-optimal performance is largely consistent with a common underlying challenge in DHOCL: different hierarchical levels exhibit inherently heterogeneous and conflicting dynamics of learning and forgetting.

This heterogeneity is most clearly exposed by the comparison with ACIL. With a frozen feature extractor, ACIL demonstrates substantially stronger knowledge retention ability at fine-grained levels, indicating that these discriminative feature for fine-grained classification are subject to more rapid forgetting due to frequent changes in the data stream. However, its pronounced degradation at coarse-

grained levels reveals a complementary limitation: coarse-level recognition requires sustained slower feature adaptation to capture shared semantics across evolving subclasses, which cannot be supported by frozen representations. From this perspective, the limitations of other anti-forgetting become more apparent. EWC++ offers negligible gains as shifting taxonomies render parameter importance a moving target, making past Fisher information an increasingly noisy and unreliable constraint. LwF suffers from even greater instability because knowledge at different hierarchical levels is learned and forgotten at different rates; a global distillation objective therefore forces the student to mimic a worse teacher (at some hierarchical levels), leading to reverse distillation. While SDC preserves structural coherence via prototype classifier, its inherently slow adaptation prevents it from tracking the rapid transitions characteristic of streaming hierarchies.

Taken together, it suggests that neither hierarchical supervision at the output level nor generic forgetting mitigation at the parameter or classifier level is sufficient on its own. Successful DHOCL calls for hierarchy-aware strategies that explicitly align feature evolution with hierarchical topology, while enabling classifiers to balance rapid adaptation to emerging nodes with long-term structural stability.

## 4. Method

Our analysis in the previous section reveals that the core challenge of DHOCL lies in the misalignment between evolving taxonomies and shifting feature spaces. Then, we propose **HALO**, which is built on two pillars: (i) a hierarchical regularization (HPR) that enforces a detailed correspondence between feature semantics and the hierarchical topology, and (ii) a balanced aggregation mechanism for the classifiers (PredLA) that reconciles rapid adaptation with long-term stability.

### 4.1. Hierarchical Prototypes Regularization (HPR)

Our first goal is to construct a flexible feature space that inherently reflects the hierarchical taxonomy. Inspired by ProtoPNet(Chen et al., 2019), we introduce a lightweight adapter $f_A$ (two $1\times1$ convolutions with a sigmoid gate) that maps backbone features into a prototype space: given $x$, the backbone yields $F = f(x) \in \mathbb{R}^{d \times H \times W}$ and the adapter outputs $M = f_A(F) \in \mathbb{R}^{d_p \times H_m \times W_m}$. For each level $h$, each class $c \in \mathcal{C}_t^h$ has a small prototype set $P_c^h = \{p_{c,j}^h \in \mathbb{R}^{d_p}\}_{j=1}^{J_c^h}$; all level-$h$ prototypes are $P^h = \{P_c^h\}_{c \in \mathcal{C}_t^h}$ with size $J_h = \sum_{c \in \mathcal{C}_t^h} J_c^h$. Index spatial positions by $i \equiv (u, v)$ and denote the feature vector at patch $i$ by $M_i \in \mathbb{R}^{d_p}$. For prototype $p_{c,j}^h$, we compute cosine similarity $s_{i,(c,j)}^h = \langle M_i, p_{c,j}^h \rangle / (\|M_i\|_2 \|p_{c,j}^h\|_2)$. Then, we aggregate similarities by a spatial max to obtain prototype scores:

$$g_{(c,j)}^h = \max_i \ s_{i,(c,j)}^h. \qquad (2)$$

Class logits are then given by a fixed class–prototype assignment mask $A^h \in \{0,1\}^{J_h \times |\mathcal{C}_t^h|}$:

$$z^h = (g^h)^\top A^h, \qquad (3)$$

where each column of $A^h$ selects the prototypes of its class.

During the training, $f$, $f_A$, and $\{P_c^h\}$ are optimized through objectives that combines Multi-CE loss with ProtoPNet-style cluster and separation costs:

$$\mathcal{L}_{\mathrm{pro}}^h = \ell_{\mathrm{ce}}(z^h, \tilde{y}^h) + \ell_{\mathrm{clst}}(P^h, M) + \ell_{\mathrm{sep}}(P^h, M). \quad (4)$$

Let $\mathcal{J}_y^h$ denote the index set of level-$h$ prototypes assigned to class $y$, and $\mathcal{J}_{\neg y}^h$ those assigned to all classes other than $y$. Then $\ell_{\mathrm{clst}}(P^h, M) = \min_{j \in \mathcal{J}_y^h} \min_i \|M_i^{(n)} - p_j^h\|_2^2$ pulls at least one patch toward a prototype of the ground-truth class, and $\ell_{\mathrm{sep}}(P^h, M) = -\min_{j \in \mathcal{J}_{\neg y}^h} \min_i \|M_i^{(n)} - p_j^h\|_2^2$ pushes patches away from non-class prototypes.

As shown in Figure 5, compared with common class-mean features, these local prototypes anchor granularity-specific key features (e.g., "wings" for birds vs."beak shape" for species). This design naturally provides flexibility for aligning features with the evolving label tree across both the hierarchy structure and time. To ensure the model's feature space evolves in tandem with the hierarchy, we introduce the following two regularizers.

(1) *Hierarchical Structural Alignment:* By encouraging higher similarity between ancestor-descendant prototype pairs, we explicitly bake the tree $\mathcal{T}_t$ into the feature space. Unlike logit-level constraints that struggle with "moving targets" in a dynamic tree, this representation-level alignment offers more flexibility while maintaining structural coherence. We use a simple margin-based ranking loss based on similarities between different prototypes to achieve this.

(2) *Temporal saliency consistency.* We penalize response changes at the most informative locations. For each prototype $p_j^h$, $\Omega_j^h$ denotes the indices of the TopK most similar patches from adapter output feautres $M$. Then, we compute the similarity $s_{i,j}^{t,h}$ between current features and current prototypes $P^t$, while $s_{i,j}^{t-1,h}$ is the similarity between current features and cached prototypes $P^{t-1}$. By constraining how current model perceives established visual concepts, this feature-level regularization provides precise supervisory signals which simultaneously ensures the stability required for coarse semantics and does not compromise the plasticity needed for new fine-grained updates, effectively mitigating the interference between heterogeneous learning dynamics

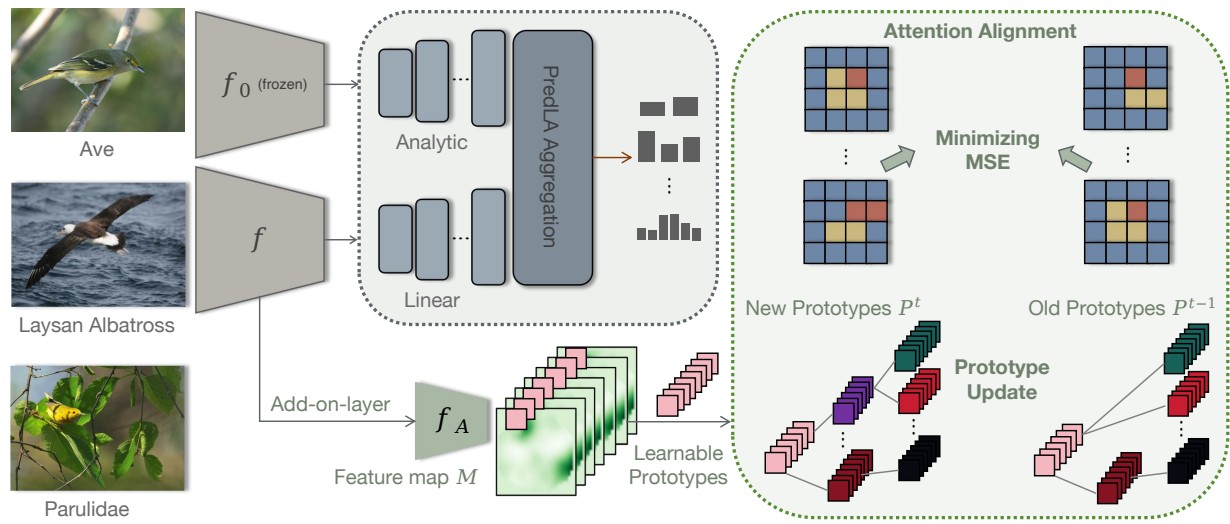

*Figure 5.* Framework overview. **Left**: a trainable backbone $f$ and a frozen backbone $f_0$ are followed by linear and analytic heads. PredLA aggregates calibrated predictions per level. **Bottom**: an add-on adapter $f_A$ maps features to a prototype space $M$. **Right**: HPR maintains class-specific prototype banks to form attention maps, enforcing consistency between consecutive steps and hierarchical alignment by pulling ancestor–descendant prototypes closer while pushing negatives outside the subtree. Together, HALO enables rapid, stable prediction, and structured feature semantics across an evolving hierarchy.

across levels. The overall HPR regularization is written as:

$$\mathcal{R} = \lambda \Big[ \sum_{(a,d)\in\mathcal{T}_t} \max\big\{0, m - S(P_a, P_d) + S(P_a, P_-)\big\}$$
$$+ \sum_{h,j} \frac{1}{K} \sum_{i\in\Omega_j^h} \big(s_{i,j}^{t,h} - s_{i,j}^{t-1,h}\big)^2 \Big],$$

(5)

where $(a, d)$ ranges over ancestor–descendant class pairs in $\mathcal{T}_t$; $s(P_1, P_2) = \max_{p\in P_1,\, q\in P_2} \cos(p, q)$ computes the maximum pairwise cosine similarity between two banks; $P_-$ is a bank from a class outside the descendant's subtree, chosen at the same hierarchy level as $d$; and $m$ is a margin. Together, these two terms stabilize semantic cores while allowing controlled drift along newly expanded branches.

## 4.2. Hierarchical Adaptive Aggregation (PredLA)

Our second goal is to resolve the *inherent conflict of learning dynamics induced by hierarchical evolution*. In DHOCL, different hierarchy levels impose fundamentally different requirements on the classifier: coarse-level decisions demand long-term adaptation under repeated taxonomy revisions, whereas fine-grained recognition must remain stable with endless newly emerging classes. Existing OCL approaches implicitly prioritize one side of this trade-off. Replay-trained linear classifiers adapt rapidly and achieve strong performance on coarse-grained classification and newly introduced fine-grained classes, but suffer from catastrophic forgetting on old fine-grained classes. Analytic classifiers (ACIL), operating on frozen representations, provide excellent stability and resistance to catastrophic forgetting,

yet lack sufficient plasticity.

Motivated by this observation, we introduce **PredLA**, a *hierarchy-aware prediction aggregation* mechanism that explicitly decomposes classification responsibility across complementary learning dynamics, without interfering with their internal optimization. Rather than merging classifiers at the representation or parameter level, PredLA operates purely at the prediction level, allowing each classifier to specialize in the regime where it is most reliable.

**Per-level complementary predictors.** Let $f(\cdot)$ denote a trainable feature extractor and $f_0(\cdot)$ a frozen pretrained backbone. At each hierarchy level $h$, we maintain two classifiers: a linear head $g_{\text{lin}}^h$ trained on $f(x)$, and an analytic (ACIL) head $g_{\text{acil}}^h$ trained on frozen features $f_0(x)$. Their corresponding logits are

$$z_\ell^h = g_{\text{lin}}^h\big(f(x)\big), \qquad z_a^h = g_{\text{acil}}^h\big(f_0(x)\big). \quad (6)$$

The linear head is optimized end-to-end to maximize adaptability, using Multi-CE loss on stream data:

$$\mathcal{L}_{\text{lin}} = \sum_{h\in\mathcal{H}_t} \ell\big(g_{\text{lin}}^h\big(f(x)\big), \tilde{y}^h\big), \quad (7)$$

where $\tilde{y}^h$ denotes the label completed along the ancestor path of $y$ in the current hierarchy $\mathcal{T}_t$. In contrast, the analytic head prioritizes stability and is updated via recursive least squares on frozen features (no backpropagation):

$$\mathcal{L}_{\text{acil}} = \sum_{h\in\mathcal{H}_t} \big\| g_{\text{acil}}^h\big(f_0(x)\big) - \tilde{y}^h \big\|_F^2 + \gamma \big\| g_{\text{acil}}^h \big\|_F^2, \quad (8)$$

*Table 1.* This table uses four datasets, each partitioned into ten sequential tasks with blurred boundaries to form the data stream. Each sample receives supervision at a randomly selected granularity as stated in Section 2. We report AAUC, FAUC, MS (lower is better), and FFAcc as defined in Section 3, providing a comprehensive comparison from multiple perspectives.

| Method | CIFAR100 | | | | FGVC-Aircraft | | | | CUB-200 | | | | iNaturalist | | | |
|---|---|---|---|---|---|---|---|---|---|---|---|---|---|---|---|---|
| | AAUC | FAUC | MS ↓ | FFAcc | AAUC | FAUC | MS ↓ | FFAcc | AAUC | FAUC | MS ↓ | FFAcc | AAUC | FAUC | MS ↓ | FFAcc |
| RS (HAF)(Garg et al., 2022) | 54.0 | 37.6 | 1.22 | 25.3 | 20.2 | 16.2 | 2.03 | 9.9 | 50.5 | 20.3 | 1.51 | 15.7 | 71.7 | 24.8 | 2.02 | 11.6 |
| CBRS(Chrysakis & Moens, 2020) | 39.9 | 27.4 | 1.53 | 9.88 | 26.4 | 24.1 | 1.87 | 11.5 | 49.8 | 19.7 | 1.45 | 18.6 | 70.4 | 26.3 | 1.90 | 14.7 |
| MIR(Aljundi et al., 2019a) | 39.5 | 23.3 | 1.89 | 12.4 | 32.1 | 23.6 | 1.82 | 12.4 | 53.5 | 27.9 | 1.42 | 21.4 | 68.4 | 23.0 | 2.27 | 15.3 |
| CLIB(Koh et al., 2022) | 52.4 | 34.6 | 1.41 | 19.5 | 23.9 | 17.0 | 1.94 | 11.7 | 50.6 | 23.4 | 1.44 | 18.9 | 72.8 | 26.4 | 1.93 | 19.4 |
| RM(Bang et al., 2021) | 52.7 | 39.0 | 1.25 | 18.8 | 31.5 | 22.6 | 1.94 | 13.4 | 52.3 | 23.1 | 1.40 | 17.4 | 72.0 | 25.4 | 2.07 | 14.4 |
| PLFMS(Lee et al., 2023) | 46.5 | 35.4 | 1.34 | 24.0 | 26.1 | 23.3 | 1.81 | 13.3 | 52.2 | 22.9 | 1.53 | 16.3 | 70.4 | 25.4 | 2.03 | 16.2 |
| iCaRL w/ SDC(Yu et al., 2020) | 49.8 | 39.6 | 1.19 | 23.1 | 28.4 | 22.7 | 1.80 | 18.1 | 47.4 | 24.5 | 1.43 | 22.4 | 67.3 | 28.1 | 1.89 | 17.1 |
| LwF w/ RS(Li & Hoiem, 2017) | 54.1 | 36.5 | 1.21 | 27.2 | 24.3 | 16.1 | 1.94 | 11.1 | 50.8 | 21.8 | 1.47 | 16.7 | 60.8 | 17.6 | 2.35 | 14.3 |
| EWC++ w/ RS(Kirkpatrick et al., 2017) | 53.3 | 34.9 | 1.39 | 30.8 | 30.9 | 15.8 | 1.93 | 10.3 | 51.5 | 23.9 | 1.46 | 14.7 | 65.4 | 21.9 | 2.32 | 18.4 |
| ICICLE w/ RS(Rymarczyk et al., 2023) | 49.0 | 30.8 | 1.37 | 19.6 | 18.4 | 9.80 | 1.77 | 7.30 | 46.4 | 21.6 | 1.40 | 15.6 | 57.4 | 20.1 | 2.39 | 9.40 |
| ACIL(Zhuang et al., 2022) | 49.5 | 39.2 | 1.37 | 35.8 | 38.1 | 27.4 | 1.77 | **30.5** | 52.5 | 30.9 | 1.34 | 35.8 | 67.2 | 34.6 | 2.24 | 26.8 |
| GACIL(Zhuang et al., 2024a) | 51.5 | 43.9 | 1.12 | **36.1** | 38.6 | 26.7 | 1.75 | 30.3 | 54.7 | 31.2 | 1.30 | 35.8 | 69.0 | 34.1 | 2.20 | 30.9 |
| OnPro(Wei et al., 2023) | 50.6 | 31.5 | 1.45 | 20.9 | 25.6 | 28.3 | 1.93 | 21.6 | 47.6 | 20.9 | 1.51 | 15.1 | 61.4 | 17.7 | 2.16 | 13.5 |
| NsCE(Wang et al., 2024b) | 53.4 | 36.7 | 1.34 | 24.0 | 26.8 | 18.4 | 1.85 | 22.3 | 50.3 | 24.6 | 1.49 | 17.7 | 69.2 | 21.4 | 2.35 | 14.3 |
| CCL-DC(Wang et al., 2024a) | 54.5 | 35.9 | 1.42 | 25.8 | 33.1 | 28.0 | 1.75 | 25.9 | 51.5 | 20.8 | 1.42 | 18.4 | 70.8 | 27.2 | 2.03 | 16.5 |
| OnLora(Wei et al., 2025) | 52.1 | 34.2 | 1.28 | 28.4 | 34.5 | 26.7 | 1.69 | 24.7 | 53.2 | 28.4 | 1.29 | 28.9 | 73.5 | 30.2 | 1.89 | 22.1 |
| Ours | **59.2** | **43.5** | **1.13** | 32.0 | **40.6** | **36.6** | **1.28** | 28.9 | **62.9** | **46.2** | **0.93** | **37.6** | **78.3** | **40.8** | **1.55** | **40.1** |

with a small regularization coefficient $\gamma > 0$. Importantly, these two predictors are trained independently, preserving their distinct learning dynamics. Thus, the overall loss for stream data can be formulated as follows:

$$\mathcal{L} = \mathcal{L}_{\text{lin}} + \mathcal{L}_{\text{pro}} + \mathcal{R}. \tag{9}$$

**Hierarchy-aware prediction aggregation.** To further combine these classifiers without sacrificing adaptability or destabilizing long-term knowledge, PredLA aggregates in probability space, allowing calibrated contributions from each head while avoiding scale dominance caused by heterogeneous objectives. Specifically, at each level $h$, we apply temperature scaling to obtain

$$p_\ell^h = \text{softmax}\big(z_\ell^h/\tau_\ell^h\big), \qquad p_a^h = \text{softmax}\big(z_a^h/\tau_a^h\big), \tag{10}$$

and compute the aggregated prediction

$$\hat{p}^h = \alpha_\ell^h \, p_\ell^h + \alpha_a^h \, p_a^h, \quad \alpha_\ell^h, \alpha_a^h \geq 0, \ \alpha_\ell^h + \alpha_a^h = 1. \tag{11}$$

**Replay-conditioned aggregation learning.** A key principle of PredLA is to decouple aggregation learning from stream updates. Optimizing aggregation weights on incoming data would entangle PredLA with short-term distribution shifts and undermine its role as a stabilizing mechanism. We therefore learn the aggregation weights $\alpha$ and temperatures $\tau$ *exclusively on replayed samples* from the memory buffer $\mathcal{M}$. Concretely, aggregation weights are exclusively opti-

mized on replayed data $\mathcal{M}$ via a bilevel objective:

$$\min_{\theta, \alpha} \ \mathbb{E}_{(x,y)\sim\mathcal{M}} \sum_{h\in\mathcal{H}_t} \ell_{\text{ce}}\big(\hat{p}^h(x;\theta,\alpha,\tau), \tilde{y}^h\big),$$

$$\min_{\tau} \ \mathbb{E}_{(x,y)\sim\mathcal{M}} \sum_{h\in\mathcal{H}_t} \max\Big\{0, \big|\mathcal{H}(p_\ell^h) - \mathcal{H}(p_a^h)\big| - \delta\Big\}. \tag{12}$$

In Eq.12, $\theta$ collects the network and classifier parameters, $\delta$ is a small tolerance, and $\mathcal{H}(p) = -\sum_c p_c \log p_c$ denotes the Shannon entropy. Note that the update for $\tau$ is performed alongside each batch. Since optimizing $\tau$ involves only a single scalar per level and incurs negligible computational overhead, we perform a fixed one-step gradient update for it in each iteration. Despite its simplicity, PredLA can successfully resolve the stability-plasticity conflict inherent in dynamic hierarchical learning. For a detailed algorithmic description, please refer to Appendix B.4.

## 5. Experiments

In this section, we follow standard hierarchical classification and OCL settings evaluated on four datasets: CIFAR-100, FGVC-Aircraft, CUB-200, and iNaturalist. Each dataset can be finally organized into a balanced label tree with uniform depth. We simulate a DHOCL scenario by partitioning fine-grained classes into 10 splits and constructing a data stream as defined in Section 2, where the stream introduces new samples, new labels, and potentially new granularities. For fair comparison, all methods use the same replay buffer size: 1,000 for CIFAR-100, 1,000 for FGVC-Aircraft, 1,000 for CUB-200, and 5,000 for iNaturalist. Unless otherwise noted, we adopt the data augmentation and optimization settings from prior works (Koh et al., 2022; Bertinetto et al.,

2020). All results reported are averaged over five random seeds. Detailed implementation details on baseline methods and datasets are provided in the Appendix A.

**Main Results.** We first compare HALO to OCL baselines adapted to DHOCL, grouping methods into replay- (Upper in the Table 1) and regularization-based categories. As shown in Table 1, on CIFAR-100, FGVC-Aircraft, CUB, and iNaturalist, HALO consistently improves AAUC, FAUC, and MS, indicating higher accuracy across granularities and lower error severity over training. The learning curves in Figure 4 further highlight its favorable stability–plasticity trade-off: relative to both replay- and regularization-based baselines, HALO learns new classes and hierarchy levels quickly while limiting forgetting. Compared with analytic classifiers such as ACIL and GACIL, HALO benefits from full-model updates and avoids failures tied to the limited discriminability of pretrained features, yielding particularly large gains at the coarse level while remaining competitive in final fine-grained accuracy.

We next study backbone and pretraining choices. As summarized in Table 3, HALO consistently outperforms ACIL and CCL (Wang et al., 2024a) across architectures and initializations—including ResNet-50 and ViT-B with Supervised, CLIP (Radford et al., 2021), DINO (Caron et al., 2021), and MAE (He et al., 2022) pretraining, indicating robustness to model capacity and pretraining variants. It's noticeable that among three methods, ACIL is the most sensitive to the backbone and pretraining because it uses frozen features. Its performance therefore closely tracks the discriminative power of the pretrained encoder, yielding higher variance across initializations. Beyond these results, we also (i) analyze buffer size, memory sampling strategy, task length, and split strategies (C.1); (ii) evaluate on ImageNet-H, whose hierarchy is naturally imbalanced with non-uniform depth (C.2); (iii) assess runtime efficiency and the computational overhead of different modules (C.3); (iv) conduct a systematic analysis of different methods under varying delays of label tree updates (C.4) (the delay is set to one iteration in the main paper). Due to space constraints, these detailed results and implementation are provided in the Appendix C.

To further evaluate the robustness of HALO against hierarchical noise, we conduct experiments on corrupted iNaturalist hierarchies subjected to edge deletions (vacant connections) and random perturbations (erroneous connections). As shown in Table 4, we observe two key insights: (1) HALO exhibits graceful degradation: even under a high vacancy rate of 40%, performance metrics decline by less than 3%. (2) Structural noise is more detrimental than incompleteness: erroneous connections impact performance more significantly than missing edges. For instance, 40% noise leads to a $-5.01\%$ drop in AAUC, whereas 40% vacancy results in only a $-2.18\%$ decrease.

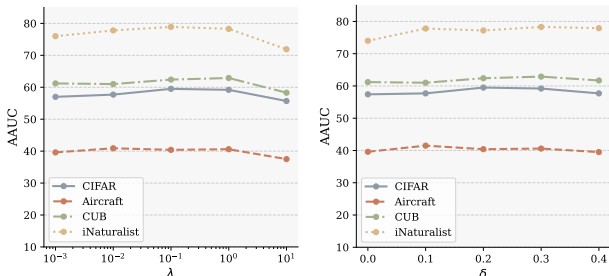

*Figure 6.* Sensitivity analysis on $\lambda$ and $\delta$.

**Ablation Study.** Table 2 reports an ablation over four datasets and four metrics. We compare shared components in our framework (Linear, Analytic, ICICLE) as well as the proposed PredLA and HPR. It can be seen that PredLA consistently boosts accuracy (AAUC/FAUC) and stability (MS), with pronounced gains on fine-grained datasets (e.g., on CUB-200, AAUC improves from 52.5 to 61.2 and MS decreases from 1.34 to 0.97). Meanwhile, incorporating HPR provides small but reliable additional gains, particularly in FFAcc (e.g., on iNaturalist, from 35.2 to 40.1).

Moreover, we also directly visualize how hierarchical prototypes structurally align with the taxonomy by visualizing learned prototype distances during training on CIFAR-100. As shown in Fig. 13, we compute the normalized pairwise distances between class prototypes in HPR's embedding space as $d_{ij} = 1 - \max_{p \in P_i, q \in P_j} \cos(p, q)$, where the similarity term matches our optimization objective in Eq. 5. We compare these learned distances against ground-truth (ideal) taxonomic distances (measured by normalized LCA depth in the hierarchy tree). The results show, as HPR is applied, the learned prototype distance matrices become increasingly aligned with the ground-truth hierarchies.

**Sensitivity Analysis.** Given that HALO integrates multiple modules, it inevitably introduces some tunable hyperparameters, such as the regularization coefficient $\gamma$ in Eq. 4, the number of prototypes per class $|P_c^h|$ in HPR, and the top-$K$ and margin $m$ in Eq. 5. Many of these either have sensitivity analyses or established settings in prior work, or show limited impact within reasonable ranges (e.g., the margin $m$). Therefore, we focus on two influential hyperparameters: the HPR coefficient $\lambda$ and the bi-level trade-off $\delta$ in Eq. 12. As shown in Fig. 6, performance remains stable as long as $\lambda$ is not too large (overly strong constraints reduce plasticity) and $\delta$ is not too small (overly strict temperature control can overflatten the linear classifier's predictions). Concrete values and justifications for all other hyperparameters are provided in the Appendix B.

*Table 2.* Ablation study of the proposed PredLA and HPR modules.

| Method | CIFAR100 | | | | FGVC-Aircraft | | | | CUB-200 | | | | iNaturalist | | | |
|---|---|---|---|---|---|---|---|---|---|---|---|---|---|---|---|---|
| | AAUC | FAUC | MS ↓ | FFAcc | AAUC | FAUC | MS ↓ | FFAcc | AAUC | FAUC | MS ↓ | FFAcc | AAUC | FAUC | MS ↓ | FFAcc |
| Linear | 54.0 | 37.6 | 1.22 | 25.3 | 20.2 | 16.2 | 2.03 | 9.9 | 50.5 | 20.3 | 1.51 | 15.7 | 71.7 | 24.8 | 2.02 | 11.6 |
| Analytic | 49.5 | 39.2 | 1.37 | 35.8 | 38.1 | 27.4 | 1.77 | 30.5 | 52.5 | 30.9 | 1.34 | 35.8 | 67.2 | 34.6 | 2.24 | 26.8 |
| ICICLE | 49.0 | 30.8 | 1.37 | 19.6 | 18.4 | 9.80 | 1.77 | 7.30 | 46.4 | 21.6 | 1.40 | 15.6 | 57.4 | 20.1 | 2.39 | 9.40 |
| PredLA | 58.4 | 42.3 | 1.16 | 29.1 | 40.4 | 35.8 | 1.33 | 24.9 | 61.2 | 44.6 | 0.97 | 35.2 | 76.4 | 39.4 | 1.58 | 35.2 |
| PredLA w/ HPR | 59.2 | 43.5 | 1.13 | 32.0 | 40.6 | 36.6 | 1.28 | 28.9 | 62.9 | 46.2 | 0.93 | 37.6 | 78.3 | 40.8 | 1.55 | 40.1 |

*Table 3.* Performance of ACIL, CCL and HALO with different backbones and pretraining methods.

| Dataset | Backbone / Pretrain | AAUC ↑ | | | MS ↓ | | |
|---|---|---|---|---|---|---|---|
| | | ACIL | CCL | Ours | ACIL | CCL | Ours |
| Aircraft | ViTB / CLIP | 30.3 | 36.9 | 40.4 | 1.89 | 1.68 | 1.28 |
| | ViTB / DINO | 34.2 | 35.7 | 41.5 | 1.85 | 1.70 | 1.24 |
| | ViTB/ / MAE | 27.9 | 33.5 | 39.7 | 2.14 | 1.73 | 1.37 |
| | Res50 / Sup | 35.5 | 34.4 | 40.2 | 1.81 | 1.72 | 1.33 |
| | Res50 / DINO | 38.1 | 33.1 | 40.6 | 1.77 | 1.75 | 1.28 |
| iNaturalist | ViTB / CLIP | 74.2 | 70.2 | 80.4 | 1.89 | 1.97 | 1.52 |
| | ViTB / DINO | 80.4 | 73.9 | 85.8 | 1.50 | 1.84 | 1.41 |
| | ViTB/ MAE | 62.1 | 71.2 | 77.4 | 2.41 | 1.93 | 1.69 |
| | Res50 / Sup | 65.4 | 68.3 | 76.0 | 2.27 | 2.10 | 1.65 |
| | Res50 / DINO | 67.2 | 70.8 | 78.3 | 2.24 | 2.03 | 1.55 |

*Table 4.* Robustness to hierarchy noise on iNaturalist. We evaluate HALO under corrupted hierarchies with edge deletions (vacant) and random perturbations (noise). Our model demonstrates graceful degradation even under significant structural corruption.

| Noise connection | Vacant connection | FFAcc ↑ | AAUC ↑ | FAUC ↑ | MS ↓ |
|---|---|---|---|---|---|
| 0% | 0% | 40.42 | 78.06 | 41.02 | 1.56 |
| 0% | 20% | 40.17 | 76.40 | 41.31 | 1.59 |
| 0% | 40% | 40.23 | 75.88 | 40.76 | 1.61 |
| 20% | 0% | 40.08 | 75.13 | 40.08 | 1.64 |
| 40% | 0% | 40.03 | 73.05 | 39.74 | 1.71 |

## 6. Conclusion

In this paper, we introduced DHOCL, a new realistic OCL setting where supervision arrives at arbitrary hierarchical levels and the taxonomy evolves over time. We analyzed why existing hierarchical classification and OCL methods struggle in this regime, and proposed HALO, which combines two complementary classification heads with learnable hierarchical prototypes to balance fast adaptation and stable consolidation. Although our setting still falls short of full real-world complexity (e.g., skewed or corrupted tree structures), we view this work as a first step. We hope this work offers the community a practical setting and benchmark, with observations that could inform future research in both hierarchical classification and continual learning.

## Impact Statement

This work advances online continual learning under evolving hierarchical taxonomies, with potential applications in biodiversity monitoring, medical diagnosis, and e-commerce categorization. Our method enables adaptive learning systems that can incorporate new knowledge without full retraining, reducing computational costs and enabling deployment in resource-constrained environments. Like all continual learning systems, our approach could be misused in surveillance applications or to perpetuate biases if trained on non-representative data. We emphasize the importance of diverse training data and ongoing monitoring.

## Acknowledgments and Disclosure of Funding

This work was supported by the National Science and Technology Major Project of China (No. 2024YFB3311401), the National Natural Science Foundation of China (NSFC) under Grant No. 62376126 and No. 62576168, and the Open Project Funds for the Joint Laboratory of Spatial Intelligent Perception and Large Model Application (SIPLMA-2024-YB-05).

This research has also received funding from the European Union's Horizon Europe research and innovation programme under grant agreement No. 101214398 (ELLIOT). Furthermore, this work was supported by the Spanish Ministry of Science and Innovation (MCIN/AEI/10.13039/501100011033) and the European Regional Development Fund (FEDER) under project PID2022-143257NB-I00, and the grant RYC2021-032765-I.

Funding was also provided by the National Science Centre (NCN, Poland) under Grant No. 2023/51/D/ST6/02846, and the Funding for Outstanding Doctoral Dissertation in NUAA (BCXJ25-21).

At last, we thank the anonymous reviewers for their valuable comments and suggestions.

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

# A. Related Works

**Hierarchical Classification.** Hierarchical classification addresses the problem of organizing and predicting multiple labels within taxonomic structures, which naturally arise in many domains such as product categorization, species classification, and disease diagnosis(Silla Jr & Freitas, 2011; Yang et al., 2018; Yurochkin et al., 2019; Sinha et al., 2024). Beyond improving prediction accuracy like traditional multi-label classification(Zhang & Zhou, 2013; Zhang & Wu, 2014; Zhang & Zhang, 2010; Yang et al., 2025), it enables multi-granularity predictions and controllable specificity in classification outputs(Stevens et al., 2024). Existing approaches can be mainly categorized into: (1) *Embedding-based methods* (Nickel & Kiela, 2017; Barz & Denzler, 2019; Yu et al., 2022; Liang & Davis, 2023) that encode hierarchical relationships through some specific label or feature embeddings (these embeddings can be typically perceived as class-mean vectors) derived from hierarchical distances or predefined semantics, then align visual features with these embeddings; (2) *Loss-based methods* that design hierarchical-aware loss functions, such as hierarchical cross-entropy (Bertinetto et al., 2020; Garg et al., 2022; Sinha et al., 2024) or optimal transport-based losses(Yang et al., 2018; Yurochkin et al., 2019), to penalize misclassifications based on their semantic distances in the hierarchy; and (3) *Architecture-based methods* (Liang & Davis, 2023; Wu et al., 2016; Liang et al., 2018) that employ multi-level classification heads or dynamic network structures to handle different granularities(Wu et al., 2016; Liang et al., 2018; Chang et al., 2021; Lu et al., 2025). While these methods effectively exploit hierarchical label structures, they are primarily designed for static settings with fixed label hierarchies. Recent attempts to extend hierarchical classification to continual learning scenarios assume labels arrive in a strictly coarse-to-fine order along predefined hierarchical paths(Abdelsalam et al., 2021; Lee et al., 2023). However, this assumption does not hold in real-world applications where data may arrive at arbitrary granularities—a fine-grained sample (e.g., "Golden Retriever") may appear before its coarse-grained ancestor (e.g., "Dog"), and the hierarchical structure itself may dynamically change or expand as new concepts emerge.

**Online Continual Learning.** Online Continual Learning (OCL) addresses the challenge of learning from an endless data stream while retaining previously acquired knowledge (Gunasekara et al., 2023). Similar to continual learning(Lai et al., 2025; Wang et al., 2025b), depending on what changes in the data stream, OCL can be categorized into three paradigms: task-incremental learning (new tasks with different objectives), domain-incremental learning (same tasks but different data distributions), and class-incremental learning (different new classes within the different tasks). Across all settings, existing OCL methods primarily focus on two aspects: (1) *Memory management*: designing effective buffer storage and sampling strategies that maintain class balance, sample diversity, and representativeness (Chaudhry et al., 2019a;b; Aljundi et al., 2019a;b; Chaudhry et al., 2020; Shim et al., 2021; Wang et al., 2022; Caccia et al., 2022; Ghunaim et al., 2023; Wang et al., 2025c;a); and (2) *Anti-forgetting feature learning*: improving feature discriminability and classifier robustness under single-pass training constraints (Rebuffi et al., 2017; Zhuang et al., 2022; Wang et al., 2024b; Zhuang et al., 2024a; Liu et al., 2025), often through techniques such as contrastive learning (Mai et al., 2021; Cha et al., 2021; Wei et al., 2023), mutual information maximization (Gu et al., 2022; Guo et al., 2022), and knowledge distillation (Wang et al., 2024a; Yan et al., 2024). Moreover, the analytic classifier-based approach, which decouples representation learning and classification by using closed-form solutions for classifier weights, has recently gained attention for its efficiency and robustness in CL (Zhuang et al., 2022; 2024a;b). It benefits from the stability of fixed classifier weights, which can mitigate forgetting and facilitate learning under single-pass constraints. However, nearly all existing OCL methods assume a flat label space where all classes exist at the same semantic level, ignoring the natural hierarchical relationships among concepts that are prevalent in real-world applications.

*Table 5.* Statistics of datasets used in our experiments. Each dataset differs in hierarchical depth and class balance, covering both artificial and naturally defined taxonomies.

| Dataset | Train / Test Samples | Levels | Data Balanced? | Tree Balanced? | Input Size |
|---|---|---|---|---|---|
| CIFAR-100 | 50,000 / 10,000 | 3 (superclass / class / subclass) | Strictly balanced | Yes | $32 \times 32$ |
| FGVC-Aircraft | 6,667 / 3,333 | 3 (family / variant / subvariant) | Approximately balanced | Yes | $224 \times 224$ |
| CUB-200 | 5,994 / 5,794 | 3 (order / family / species) | Approximately balanced | Yes | $224 \times 224$ |
| iNaturalist-2018 | 437,513 / 24,426 | 7 (kingdom $\rightarrow$ species) | Highly imbalanced | Yes | $224 \times 224$ |
| ImageNet-H | 1,281,167 / 50,000 | 11 (WordNet hierarchy) | Approximately balanced | No | $224 \times 224$ |

# B. Dataset Construction and Implementation Details

## B.1. Dataset Construction Details

**Hierarchical Structures**   In the main text, we adopt four standard hierarchical classification benchmarks to instantiate our DHOCL setting: CIFAR-100, FGVC-Aircraft, CUB-200, and iNaturalist. We will provide the finalized hierarchical structures for each dataset as `.pkl` files in the supplementary materials to facilitate reproducibility. In Table 5, we summarize key statistics of these datasets, including the number of training and test samples, the number of hierarchy levels, data balance, tree balance, and input image size. Notably, while CIFAR-100, FGVC-Aircraft, and CUB-200 have complete and balanced hierarchies, iNaturalist exhibits a highly imbalanced class distribution despite having a balanced tree structure. And ImageNet-H, which we discuss later, has an imbalanced hierarchy tree.

To construct the DHOCL data stream described in the main text, we follow the procedure outlined in Section 2, which produces a gradual, boundary-free stream through fine-grained class overlap. Specifically, we first randomly partition the fine-grained classes into several groups, each containing a subset of classes, and then introduce overlap between consecutive groups to blur boundaries and avoid explicit task switches. This partitioning is, of course, not unique. In Appendix C, we further evaluate alternative strategies—such as coarse-grained partitioning—and report the corresponding results. In the coarse-grained setting, classes are grouped by higher-level categories defined by the hierarchy, and the stream is formed by overlapping these coarse groups. For experiments involving a larger number of groups, we divide the fine-grained classes into more partitions to increase the diversity and complexity of the evolving stream. The corresponding results are summarized in Table 7. Moreover, for datasets such as FGVC-Aircraft, CUB-200, and iNaturalist, some fine-grained classes contain relatively few samples, and the partial hierarchical supervision further increases the learning difficulty. Consequently, adhering to a strict one-epoch training protocol can cause severe underfitting. To mitigate this issue, we duplicate the data stream with random augmentations—twice for FGVC-Aircraft and CUB-200, and four times for iNaturalist—ensuring that each sample is observed only once in its augmented form, potentially with different hierarchical labels.

**Imbalanced Hierarchies**   Beyond the datasets used in the main text—whose final label trees are complete and balanced—we also experiment on ImageNet-H, which has an imbalanced hierarchy. We first explain how to construct a DHOCL stream under such imbalance. The overall idea mirrors the main text: partition classes into several groups and blur boundaries by introducing class overlap between consecutive groups. The key difference is ambiguity caused by unequal depths; for example, one fine-grained class may be at depth 7 from the root while another is at depth 11. Although these classes share the same finest-level granularity, their ancestor nodes do not align cleanly across levels, so we cannot assign them to a common "semantic level" unambiguously. To avoid this, we prioritize partitioning at the finest level and then complete ancestors with a randomized padding strategy: we duplicate selected ancestor nodes as intermediate placeholders so that, as the stream progresses, the imbalanced tree is maintained in a temporarily balanced form, eliminating alignment ambiguities. We present the corresponding results in Table 9.

## B.2. Implementation Details

In this section, we provide implementation details of our proposed HALO method, including the both training details of the both two heads with PredLA and the hierarchical prototype regularization. At the end, we also present the algorithmic flow of HALO in Algorithm 1.

**Training of linear heads.**   Training the linear heads is straightforward: during training we directly optimize both the linear heads and the feature extractor with the cross-entropy loss. Given a mini-batch of features and one-hot labels, $\Phi = f(X) \in \mathbb{R}^{d \times b}$ and $Y \in \mathbb{R}^{C \times b}$, we minimize the following cross-entropy objective:

$$\mathcal{L}_{\mathrm{lin}} = \sum_{h \in \mathcal{H}_t} \ell_{\mathrm{ce}}\big(g_{\mathrm{lin}}^h(f(x)),\, \tilde{y}^h\big), \tag{13}$$

During replay, however, we adopt a hierarchy-aware objective that (i) trains each head against its own targets at level $h$, and (ii) enforces cross-level consistency by matching the level-$h$ prediction with the aggregation of level-$(h{+}1)$ predictions pushed up to level $h$.

$$\mathcal{L}_{\mathrm{lin}} = \sum_{h \in \mathcal{H}_t} \Big[\ell_{\mathrm{ce}}\big(g_{\mathrm{lin}}^h(f(x)),\, \tilde{y}^h\big) \;+\; \tfrac{1}{2}\,\mathrm{JS}\big(p^h \,\|\, \hat{p}^h\big)\Big], \tag{14}$$

where $p^h = \text{softmax}(g^h_{\text{lin}}(f(x)))$ is the level-$h$ predictive distribution. The term $\hat{p}^h$ is obtained by aggregating the level-$(h+1)$ distribution $p^{h+1}$ upward according to the hierarchy $T_t$; concretely, for any class $c$ at level $h$ with child set $\mathcal{C}(c)$ at level $h+1$,

$$\hat{p}^h(c) = \sum_{c' \in \mathcal{C}(c)} p^{h+1}(c'). \tag{15}$$

Equation (14) thus combines a level-wise supervised loss $\ell_{\text{haf}}$ with a Jensen–Shannon divergence that encourages the level-$h$ head to be consistent with the upward-aggregated prediction from level $h+1$. The label tree $T_t$ supplies the parent–child relations used in the aggregation (15).

**Training of analytic heads.**  We optimize the analytic head using recursive least squares (RLS) as in (Zhuang et al., 2022). Given a mini-batch of features and one-hot targets, $\Phi = f(X) \in \mathbb{R}^{d \times b}$ and $Y \in \mathbb{R}^{C \times b}$, we minimize the regularized least-squares loss in Eq. 8 with ridge $\gamma > 0$. Let $W \in \mathbb{R}^{C \times d}$ denote the analytic classifier weights and let $R \in \mathbb{R}^{d \times d}$ be the regularized feature auto-correlation inverse (RFAuM), i.e., the inverse of the accumulated $\sum \Phi\Phi^\top + \gamma I_d$ up to the current step. For the incoming batch $(\Phi, Y)$, the RLS recursion updates $W$ and $R$ as follows:

$$K = R\Phi \left(I_b + \Phi^\top R\Phi\right)^{-1} \tag{16}$$

$$W \leftarrow W - (Y - W\Phi) K^\top \tag{17}$$

$$R \leftarrow R - R\Phi \left(I_b + \Phi^\top R\Phi\right)^{-1} \Phi^\top R. \tag{18}$$

Here $b$ denotes the batch size and $I_b$ the $b \times b$ identity. For initialization, set $W = 0$ and $R = \gamma^{-1} I_d$ at the start of training. When the label space expands (e.g., new classes arrive), enlarge $W$ by appending zero-initialized rows, while keeping $R$ as a $d \times d$ matrix since it is defined in feature space. If multiple analytic heads $g^h_{\text{acil}}$ are used for $h \in \mathcal{H}_t$ as in Eq. 8, apply the same recursion independently to each head with its corresponding targets $\tilde{y}^h$.

Beyond the plasticity limitation of analytic classification heads under frozen features discussed in the main text, there is a second caveat specific to the DHOCL setting. Although we can instantiate an independent analytic head for each hierarchy level, it is difficult to encode cross-class hierarchical relations within these heads under RLS. The RLS update relies on one-hot targets, whereas classes across levels are linked by nontrivial parent–child relations in a hierarchical label space; pure one-hot coding does not capture these dependencies. This aligns with our later observations: while analytic heads perform well at handling fine-grained splits and retaining old knowledge, their performance is noticeably weaker at coarser granularities.

**Regularization with HPR**  In the main text, we already provide a relatively detailed introduction to HPR. Here we add one clarification: unlike prior work such as (Rymarczyk et al., 2023; Chen et al., 2019), HPR is not trained in two separate stages. Instead, it is optimized jointly with the main task loss, while the final linear layer remains fixed and is not updated during training. Concretely, at each training step we first compute the main task loss (cross-entropy), then compute the HPR regularization term, and finally take a weighted sum of the two as the overall training objective. This joint optimization encourages the feature extractor to learn new tasks while retaining representations useful for previous tasks, thereby alleviating catastrophic forgetting.

### B.3. Hyperparameters Selection

As elaborated in the main paper, HALO integrates multiple modules, which inevitably introduces several tunable hyperparameters that may affect its performance. In this section, we provide additional details regarding the selection of these hyperparameters in our experiments. Beyond the sensitivity analysis already discussed (specifically for the HPR regularization coefficients $\lambda$ and $\delta$), we further tune the following hyperparameters: the regularization coefficient $\gamma$ in Eq.4, the number of prototypes per class $|P^h_c|$ in HPR, and the top-$K$ and margin $m$ in Eq.5. As shown in Figures 8, HALO demonstrates robust performance across a wide range of these hyperparameters under DHOCL. Moreover, most hyperparameter ranges align closely with their default values specified in the main paper, indicating the reasonableness of our default choices. Specifically, across all datasets and experiments, we adopted the following default values: $\gamma = 1.0$, $|P^h_c| = 5$, and $m = 0.1$. For $K$, consistent with the findings of (Rymarczyk et al., 2023), which suggest distilling similarity only from the most critical patches due to variations in pixel and patch numbers across different inputs, we adopt the top-10% of patches as the default for similarity distillation.

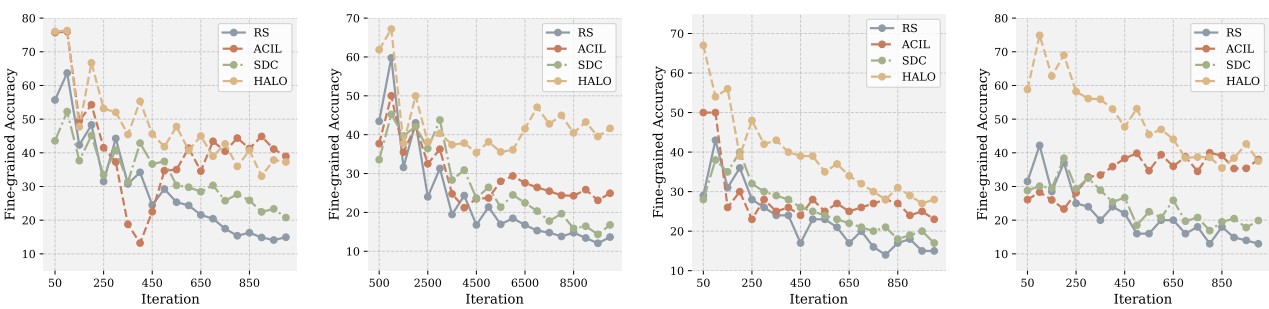

*Figure 7.* Learning dynamic of different methods on CIFAR-100, FGVC-Aircraft, CUB and iNaturalist under DHOCL. Performance is evaluated on fine-grained Acc, which reflects forgetting resistance.

*Table 6.* Performance comparison of different methods under a DHOCL stream constructed via coarse-grained partitioning.

| | CIFAR100 | | | | FGVC-Aircraft | | | | CUB-200 | | | |
| Method | AAUC | FAUC | MS↓ | FFAcc | AAUC | FAUC | MS↓ | FFAcc | AAUC | FAUC | MS↓ | FFAcc |
|---|---|---|---|---|---|---|---|---|---|---|---|---|
| RS(HAF) | 43.7 | 25.8 | 1.57 | 22.8 | 26.3 | 9.59 | 2.15 | 4.72 | 49.1 | 14.2 | 1.52 | 7.83 |
| NsCE | 51.6 | 34.4 | 1.34 | 29.1 | 30.4 | 16.4 | 1.93 | 17.6 | 53.5 | 26.4 | 1.36 | 15.9 |
| CCL-DC | 52.8 | 36.5 | 1.25 | 34.0 | 28.4 | 16.0 | 1.94 | 14.9 | 52.8 | 25.3 | 1.40 | 14.2 |
| Analytic | 53.5 | 38.2 | 1.23 | 41.0 | 37.8 | 22.6 | 1.80 | 27.8 | 60.9 | 33.2 | 1.18 | 32.2 |
| HALO | 55.7 | 40.6 | 1.17 | 36.4 | 36.7 | 24.1 | 1.61 | 20.7 | 61.1 | 35.1 | 1.06 | 37.5 |

For shared hyperparameters (learning rate, batch size, and memory buffer size), we keep settings consistent across all methods to ensure a fair comparison. We use AdamW with a learning rate of 5e-4, weight decay of 1e-4, and a batch size of 32. The memory batch size is 16 for CIFAR-100. For the other datasets, we use a batch size of 8 for stream data and 4 for memory. Memory buffer sizes follow common OCL practice: 1,000 for CIFAR-100, FGVC-Aircraft, and CUB-200, and 5,000 for iNaturalist and ImageNet. Full per-method configurations are provided in the accompanying .yaml files in the supplementary materials.

### B.4. Algorithmic Flow of HALO

The overall training procedure of HALO is summarized in Algorithm 1.

---
**Algorithm 1** Training procedure of HALO
---
1: **Input:** Data stream $\mathcal{D}_t$; Updated label tree $\mathcal{T}_t$; Current backbone $f_t$; Prototype adapter $f_A$; Prototype banks $\{P\}$; Replay buffer $\mathcal{M}$
2: Initialize prototypes for new classes; Cache pretrained backbone as $f_0$;
3: **for** $(x, y)$ from $\mathcal{D}_t$ and $(x_m, y_m)$ from $\mathcal{M}$ **do**
4:     Complete the label of stream $y$, and label of memory $y_m$ to $\tilde{y}$, $\tilde{y}_m$ along ancestor path in current tree $\mathcal{T}_t$
5:     Update the ACIL classifier independently with Eq.8 (Please refer to Appendix B.2 for details)
6:     Compute the loss on linear classifier as Eq.7
7:     Compute per-level prototype learning loss (Eq.4) with the regularization term (Eq.5): $\sum_h \mathcal{L}_{\text{pro}}^h + \mathcal{R}$
8:     Update backbone $f$, feauture adapter $f_A$, learnable prototypes $P$ and linear heads $w_{\text{lin}}$ with total loss: $\mathcal{L}_{\text{lin}} + \sum_h \mathcal{L}_{\text{pro}} + \mathcal{R}$
9:     Update $\alpha$ and $\tau$ on replay data based on bi-level objective (Eq.12)
10:     Update memory $\mathcal{M}$; Periodically cache $\{P^t\}$
11: **end for**

---

## C. Additional Experimental Results

### C.1. Different Split Strategies for DHOCL

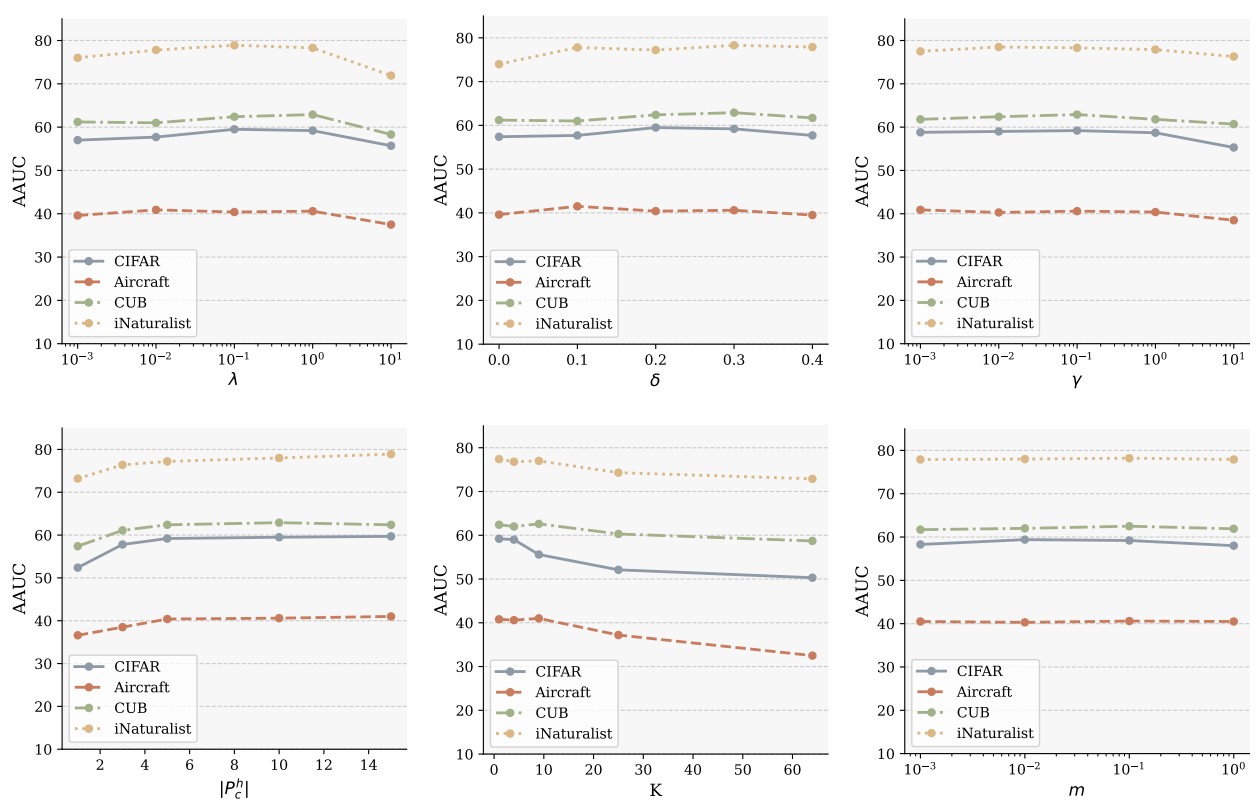

*Figure 8.* Sensitivity analysis on $\lambda$, $\delta$, $\gamma$, $|P_c^h|$, $K$, and $m$.

**Coarse-grained Split** In this section, we report results for five methods under a DHOCL stream constructed via coarse-grained partitioning, as described in Appendix B. Concretely, for CIFAR, FGVC, and CUB, we partition classes at the second granularity level into 10 groups (whereas the main paper uses the third, finest level to form 10 groups), and then blur task boundaries following iBlurry (Lee et al., 2023). The results are summarized in Table 6. While absolute performance differs from the fine-grained setting, the relative ordering of methods is consistent, and our proposed HALO achieves the best overall performance across all metrics. Notably, compared with fine-grained partitioning, the coarse-grained split tends to exacerbate forgetting for methods that do not use analytic (closed-form) classifiers. Under random fine-grained splits, models repeatedly encounter the same coarse categories across groups, which partially mitigates forgetting at the coarse level and helps stabilize feature learning. This effect is much less pronounced for HALO and the Analytic baseline, since analytic classifiers do not rely on feature stability to the same extent.

*Table 7.* Performance comparison of different methods under a DHOCL stream constructed with more split groups.

| Method | CIFAR100 | | | | Aircraft | | | | CUB-200 | | | | iNaturalist | | | |
|---|---|---|---|---|---|---|---|---|---|---|---|---|---|---|---|---|
| | AAUC | FAUC | MS ↓ | FFAcc | AAUC | FAUC | MS ↓ | FFAcc | AAUC | FAUC | MS ↓ | FFAcc | AAUC | FAUC | MS ↓ | FFAcc |
| HALO (5) | 61.2 | 51.1 | 1.02 | 45.8 | 41.5 | 37.6 | 1.25 | 30.8 | 66.7 | 48.5 | 0.92 | 41.0 | 82.4 | 47.2 | 1.41 | 43.6 |
| HALO (10) | 59.2 | 43.5 | 1.13 | 32.0 | 40.6 | 36.6 | 1.28 | 28.9 | 62.9 | 46.2 | 0.93 | 37.6 | 78.3 | 40.8 | 1.55 | 40.1 |
| HALO (50) | 55.1 | 45.6 | 1.19 | 41.5 | 37.6 | 36.1 | 1.44 | 18.4 | 64.2 | 46.5 | 0.99 | 38.2 | 80.0 | 53.7 | 1.38 | 42.6 |
| Analytic (5) | 47.7 | 39.6 | 1.41 | 45.5 | 37.3 | 29.6 | 1.78 | 30.8 | 59.1 | 35.9 | 1.15 | 37.8 | 65.4 | 27.3 | 2.46 | 23.1 |
| Analytic (10) | 49.5 | 39.2 | 1.37 | 35.8 | 38.1 | 27.4 | 1.77 | 30.5 | 52.5 | 30.9 | 1.34 | 35.8 | 67.2 | 34.6 | 2.24 | 26.8 |
| Analytic (50) | 52.6 | 42.5 | 1.30 | 42.2 | 41.1 | 27.0 | 1.70 | 27.4 | 63.1 | 44.3 | 1.04 | 36.2 | 65.4 | 29.8 | 2.52 | 20.2 |
| RS(HAF) (5) | 57.8 | 42.7 | 1.20 | 31.8 | 27.7 | 18.4 | 2.04 | 10.7 | 51.5 | 22.1 | 1.41 | 19.6 | 76.4 | 31.8 | 1.67 | 22.7 |
| RS(HAF) (10) | 54.0 | 37.6 | 1.22 | 25.3 | 20.2 | 16.2 | 2.03 | 9.9 | 50.5 | 20.3 | 1.51 | 15.7 | 71.7 | 24.8 | 2.02 | 11.6 |
| RS(HAF) (50) | 43.0 | 30.1 | 1.90 | 18.5 | 17.9 | 7.85 | 2.36 | 3.06 | 40.8 | 10.7 | 1.74 | 4.82 | 60.9 | 11.1 | 2.82 | 5.23 |

*Table 8.* Performance with different replay buffer sizes.

| Buffer Size | CIFAR100 | | | | FGVC-Aircraft | | | | CUB-200 | | | |
|---|---|---|---|---|---|---|---|---|---|---|---|---|
| | AAUC | FAUC | MS ↓ | FFAcc | AAUC | FAUC | MS ↓ | FFAcc | AAUC | FAUC | MS ↓ | FFAcc |
| HALO-100 | 52.9 | 39.6 | 1.28 | 27.8 | 39.2 | 36.0 | 1.21 | 24.5 | 58.6 | 38.7 | 1.10 | 28.1 |
| HALO-500 | 58.7 | 42.3 | 1.15 | 31.4 | 39.8 | 36.7 | 1.29 | 27.6 | 61.7 | 44.4 | 0.96 | 27.2 |
| HALO-1000 | 59.2 | 43.5 | 1.13 | 32.0 | 40.6 | 36.6 | 1.28 | 28.9 | 62.9 | 46.2 | 0.93 | 37.6 |
| HALO-5000 | 62.5 | 50.4 | 0.98 | 42.7 | 42.3 | 38.4 | 1.24 | 30.1 | 63.3 | 47.3 | 0.92 | 38.5 |

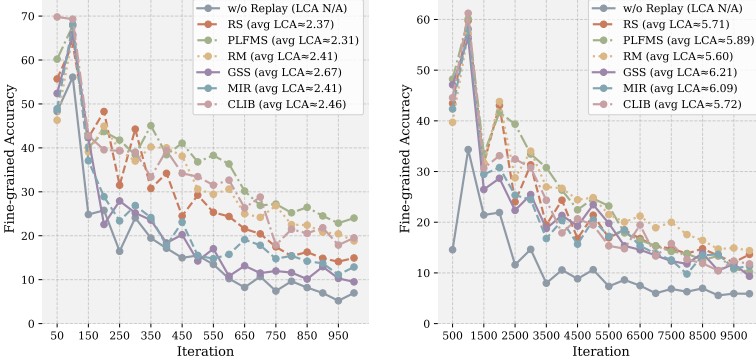

*Figure 9.* Comparison of memory sampling strategies on CIFAR-100 (left) and iNaturalist (right) under hierarchical label streams. Performance is evaluated by fine-grained accuracy, which best reflects forgetting resistance.

**More Split Groups**   In Table 7, we report results for three methods under DHOCL streams constructed with 5, 10, and 50 split groups. The overall trends are consistent with those in the main text (10 groups): increasing the number of groups generally degrades performance for all methods, as it amplifies distribution shift and reduces the frequency of revisiting prior classes.

In terms of specific methods, analytic classifiers exhibit relatively stable performance across different group counts. In contrast, fine-tuning–based methods (e.g., RS(HAF)) perform substantially better than analytic classifiers when the number of groups is small, but their performance drops markedly as the number increases. Our proposed HALO combines the strengths of fine-tuning and analytic classifiers, yielding comparatively stable and consistently superior performance across all split configurations.

**Size of Replay Buffer**   In Table 8, we report the performance of our proposed HALO method under different replay buffer sizes (100, 500, 1000, and 5000). As expected, larger buffer sizes lead to improved performance across all metrics, as they allow for more comprehensive retention of past knowledge and better mitigation of catastrophic forgetting. Notably, even with a relatively small buffer size of 100, HALO maintains competitive performance, demonstrating its effectiveness in leveraging hierarchical supervision for continual learning.

**Memory Sampling Strategy**   For a long time in the community of OCL, despite memory replay has been the de facto approach. However, our investigation of memory storage and sampling strategies under DHOCL identifies no uniformly dominant strategy, as shown in Figure 9. For this counter-intuitive finding, we give a more detailed analysis in the Sec D.

### C.2. Results on Imbalanced Hierarchy: ImageNet-H

In this section, we report results on ImageNet-H, which has an imbalanced hierarchical structure. We construct two DHOCL streams by partitioning the fine-grained classes into 10 and 20 groups, respectively, following the procedure in Appendix B. Because ImageNet is used as the downstream task, we replace the ImageNet-1k–pretrained backbone with one pretrained on iNaturalist-2018 to ensure a fairer comparison. The results are summarized in Table 9. Our proposed HALO achieves strong performance under both split settings, demonstrating its effectiveness for imbalanced hierarchies within the DHOCL framework. As discussed in the main text, our protocol reorganizes the imbalanced ImageNet-H tree by padding

*Table 9.* Results on ImageNet under two DHOCL split settings (Split-10 and Split-20).

| Method | ImageNet (Split-10) | | | | ImageNet (Split-20) | | | |
|---|---|---|---|---|---|---|---|---|
| | AAUC | FAUC | MS ↓ | FFAcc | AAUC | FAUC | MS ↓ | FFAcc |
| RS(HAF) | 53.9 | 27.4 | 6.31 | 10.4 | 47.8 | 22.3 | 7.12 | 7.85 |
| Analytic | 65.2 | 50.1 | 3.78 | 49.3 | 61.5 | 43.8 | 4.12 | 42.7 |
| NsCE | 59.8 | 36.5 | 4.29 | 34.7 | 55.3 | 32.1 | 4.56 | 30.2 |
| CCL-DC | 60.5 | 38.2 | 4.15 | 36.1 | 56.7 | 34.5 | 4.48 | 31.5 |
| HALO | **69.8** | **54.0** | **3.36** | **53.8** | **65.4** | **46.7** | **3.67** | **46.2** |

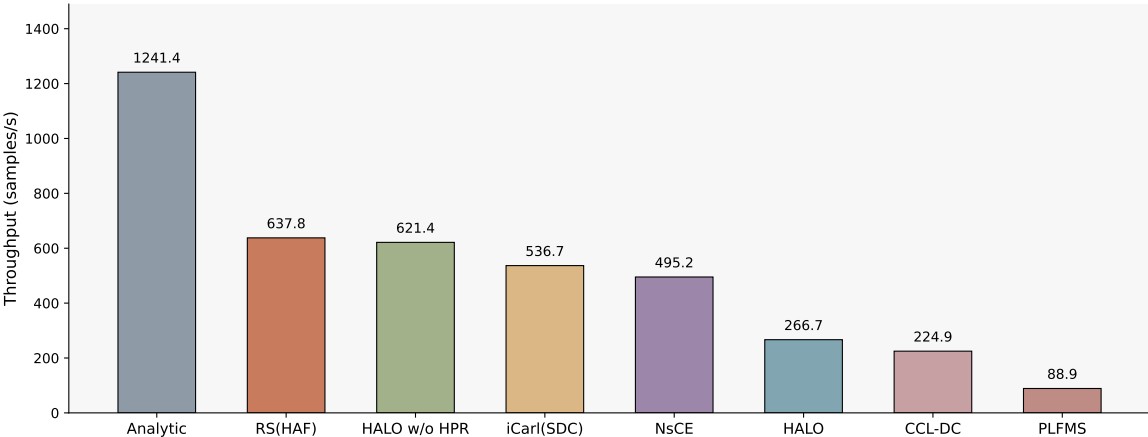

*Figure 10.* Model throughput of different methods under DHOCL on CIFAR-100. We measure the number of samples processed per second during training.

intermediate nodes, which inevitably leads to comparisons across categories at slightly different semantic granularities during evaluation. Under current classification setups, this limitation is difficult to avoid. While alternative paradigms such as VLM-based classification or QA-style evaluation may seem promising, they introduce new challenges for DHOCL problem settings: (i) VLMs' implicit understanding of hierarchical structures is difficult to quantify or control, making fair evaluation problematic; (ii) textual class labels often contain implicit (and potentially inaccurate) hierarchical information that can confound experimental design; (iii) establishing reproducible protocols for dynamic hierarchy evolution in VLM contexts remains an open problem. We believe addressing these fundamental evaluation challenges is critical future work that extends beyond the scope of this paper.

### C.3. Model Throughput

Here, we visualize training throughput (samples per second) (Zhou et al., 2022) for different methods under DHOCL on CIFAR-100 in Figure 10. As shown, HALO achieves competitive throughput relative to recent baselines, indicating strong efficiency in the DHOCL setting. Analytic methods attain the highest throughput due to closed-form updates, whereas fine-tuning–based methods are slower because of gradient computations. We also ablate HALO's PredLA module and observe a substantial throughput increase, suggesting a practical trade-off when efficiency is the priority. While our primary aim is to advance the DHOCL setting, we recognize the importance of efficiency in real-world deployments and will pursue further optimizations.

### C.4. Delay of the label hierarchy

In the main paper, we assume that the label hierarchy is updated at one iteration after the new classes arrive, which means the model will use the previous label hierarchy to learn the newly arrived classes for the current iteration (one batch of data). And after that, the label hierarchy is updated to include the newly arrived classes for the next iteration. Here, we further investigate the impact of different delay settings of the label hierarchy on model performance. Specifically, we consider three delay settings: (1) 1-iteration delay, where the label hierarchy is updated one iteration after new classes arrive (the

*Table 10.* HALO's performance with different label hierarchy delay settings.

| Delay Setting | CIFAR100 | | | | iNaturalist | | | |
|---|---|---|---|---|---|---|---|---|
| | AAUC | FAUC | MS↓ | FFAcc | AAUC | FAUC | MS↓ | FFAcc |
| 1-iteration delay | 59.2 | 43.5 | 1.13 | 32.0 | 78.3 | 40.8 | 1.55 | 40.1 |
| 10-iteration delay | 58.7 | 42.3 | 1.17 | 31.4 | 78.5 | 37.7 | 1.58 | 41.5 |
| 50-iteration delay | 56.3 | 40.4 | 1.28 | 31.2 | 76.3 | 35.8 | 1.76 | 40.3 |

default setting in the main paper); (2) 10-iteration delay, where the label hierarchy is updated five iterations after new classes arrive; (3) 50-iteration delay, where the label hierarchy is updated fifty iterations after new classes arrive. As shown in Table 10, as the delay increases, the performance of the model gradually degrades (especially the mistake severity). This is because a longer delay means that the model has to learn new classes with an outdated label hierarchy for more iterations, which may lead to more confusion and forgetting. Despite this, HALO still maintains strong performance even with a 50-iteration delay, demonstrating its robustnes.

### C.5. Hierarchical structure evolution with HPR

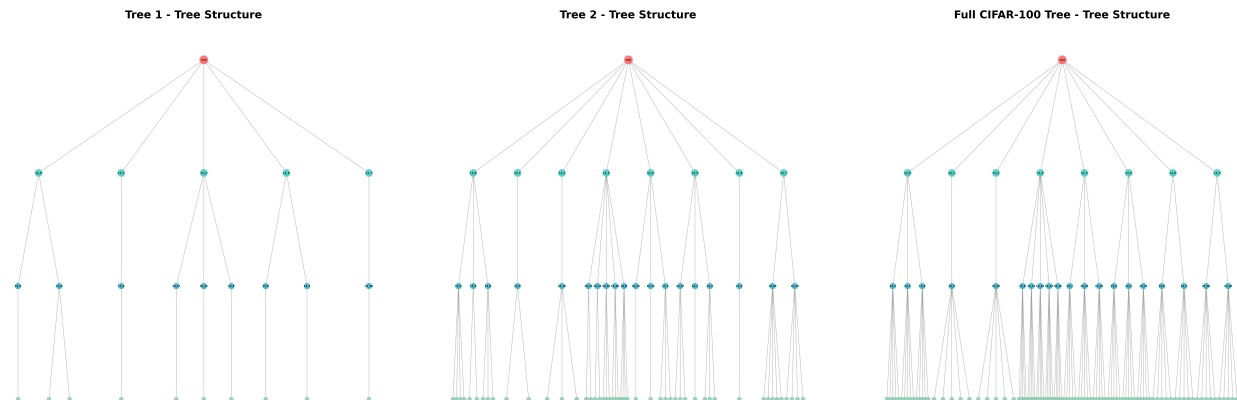

*Figure 11.* Growing label hierarchies for CIFAR-100. The left, middle, and right subfigures show the label hierarchies after confronting 10, 50, 100 (all) fine-grained classes.

As shown in Fig. 11, the label hierarchy evolves as the data stream unfolds, with new classes being added and intermediate nodes being padded to maintain a consistent tree structure. To verify that our proposed HPR module effectively captures and preserves the hierarchical relations among classes, we visualize the learned prototype hierarchies at different stages of the stream (after encountering 10, 50, and all 100 fine-grained classes) in Fig. 12. We compute distance matrices based on the ground-truth label hierarchy (using normalized LCA depth) and compare them with distance matrices derived from the learned prototypes, both with and without HPR. The results show that as HPR is applied, the learned prototype distances become increasingly aligned with the ground-truth hierarchical structure. This alignment is further quantified in Fig. 13, which visualizes the difference between the learned prototype hierarchies and the ground-truth hierarchy. The incorporation of HPR significantly reduces these discrepancies, providing direct empirical evidence that our proposed module effectively captures and preserves hierarchical relations in feature space.

### C.6. Learning Curve Analysis

In this section, we present the learning curves of different methods on all four datasets CIFAR-100, FGVC-Aircraft, CUB-200, and iNaturalist under the DHOCL setting in Figure 7. The performance is evaluated based on fine-grained accuracy (FFAcc), which reflects the model's ability to retain knowledge of previously learned fine-grained classes. As shown in Figure 7, our proposed HALO method consistently outperforms other baselines across all datasets, demonstrating its effectiveness in both rapid learning and mitigating catastrophic forgetting in the DHOCL setting.

# D. Furhter Analysis on Memory Sampling Strategies

## D.1. Detailed Analysis of Sampling Biases under Hierarchical Supervision

The limitations of the three mainstream memory-selection strategies—loss-, gradient-, and class-balanced sampling—can be further understood from the perspective of hierarchical annotation consistency.

**Loss-based sampling**   In hierarchical continual learning, each instance is associated with a label at only one hierarchy level, leading to incomplete supervision signals. As the current mini-batch typically corresponds to a single granularity, the loss function becomes dominated by that level's objectives, causing the update direction to align with the active granularity of the current step. Consequently, loss-based selection tends to overemphasize recently introduced fine-grained samples, where losses are naturally higher, and ignores the coarse-level or previously encountered classes. This short-sighted update pattern makes loss-driven replay unreliable when label granularities are mixed within the buffer.

**Gradient-based sampling**   Gradient-diversity methods, which select samples that maximize the variability of gradient directions, implicitly favor examples producing distinct high-level responses. Under hierarchical supervision, this results in a bias toward samples diverse at the coarse level, since inter-class gradients between different high-level categories dominate the gradient space. In contrast, instances belonging to the same coarse parent but differing at fine granularity—often those most challenging to retain—receive disproportionately low priority, leading to insufficient fine-level coverage during replay.

**Class-balanced sampling**   Balancing becomes more ambiguous in a hierarchical label space: it is inherently unclear which granularity level should be balanced—the coarse categories, the fine categories, or a mixture of both. Furthermore, due to incomplete labeling, certain subclasses may appear only at specific stages. When treated as minority classes, these samples can remain overrepresented in the replay memory without proper cross-level context, accelerating overfitting. As a result, the balance strategy fails to regularize effectively across hierarchy levels.

**Summary.**   Overall, all three strategies exhibit biases rooted in the uneven and incomplete granularity representation of hierarchical data. Loss-based approaches suffer from short-sighted updates, gradient-based ones from hierarchy-level bias, and class-balanced ones from ambiguous objectives. These effects collectively hinder balanced and informative replay, limiting their ability to mitigate catastrophic forgetting under hierarchical continual learning.

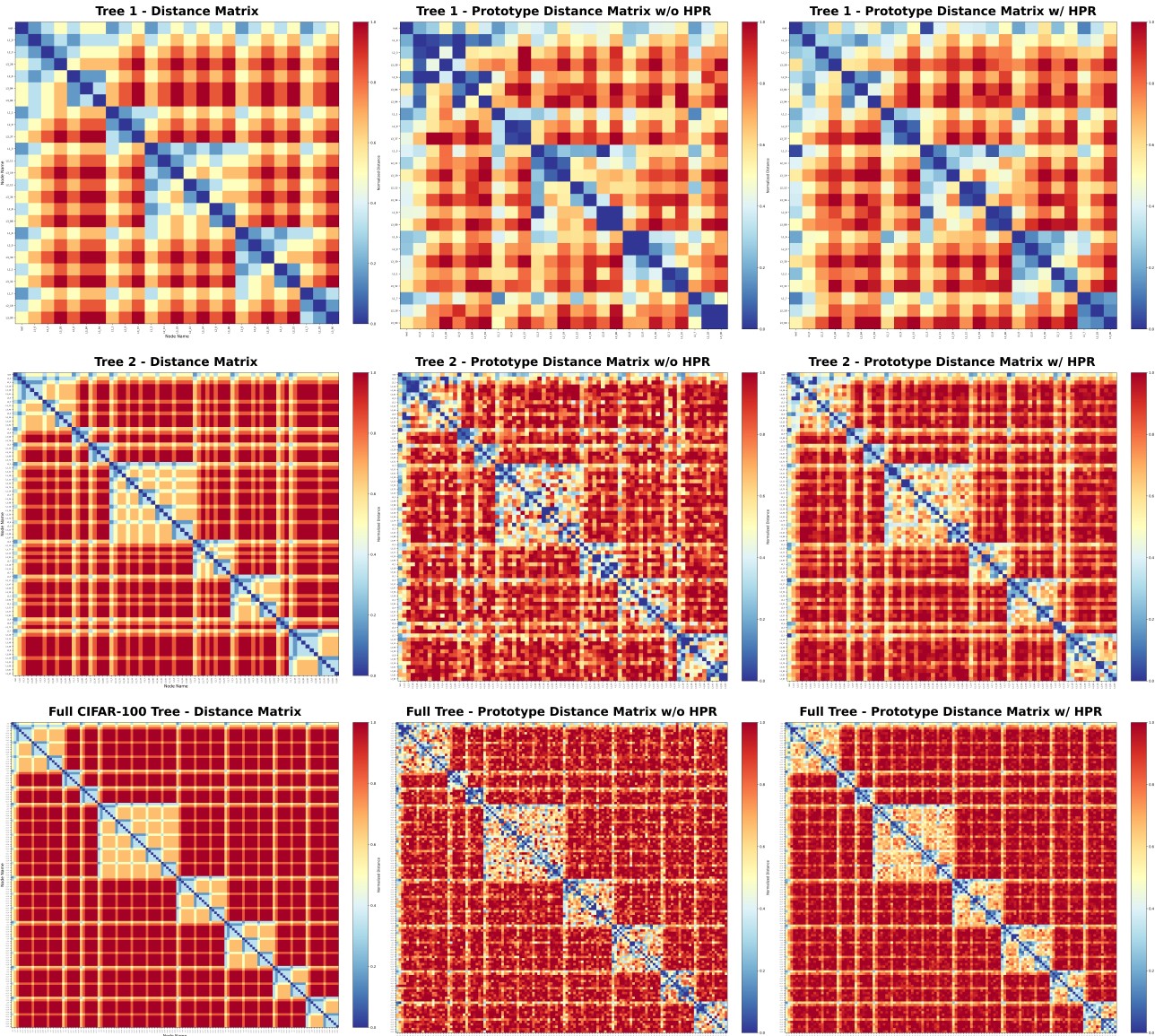

*Figure 12.* Normalized distance matrices illustrating the hierarchical relations among class labels and their prototypes. From left to right, the matrices are (i) distances derived from the ground-truth label hierarchy based on the normalized LCA (Lowest Common Ancestor) depth, (ii) distances between hierarchical prototypes obtained without hierarchical prototype regularization (HPR), and (iii) distances between hierarchical prototypes obtained with HPR. For the prototype-based matrices, each entry is computed as $d_{ij} = 1 - \max_{p \in P_i, q \in P_j} \cos(p, q)$, where $P_i$ and $P_j$ denote the prototype sets of classes $i$ and $j$. From top to bottom, "Tree 1" and "Tree 2" correspond to the scenarios in Fig. 1 where the data stream has encountered only 10 and 50 fine-grained classes, respectively, while the bottom row shows the full CIFAR100 hierarchy after all classes have been introduced. As HPR is applied, the learned prototype distance matrices become increasingly aligned with the ground-truth hierarchical distance structure.

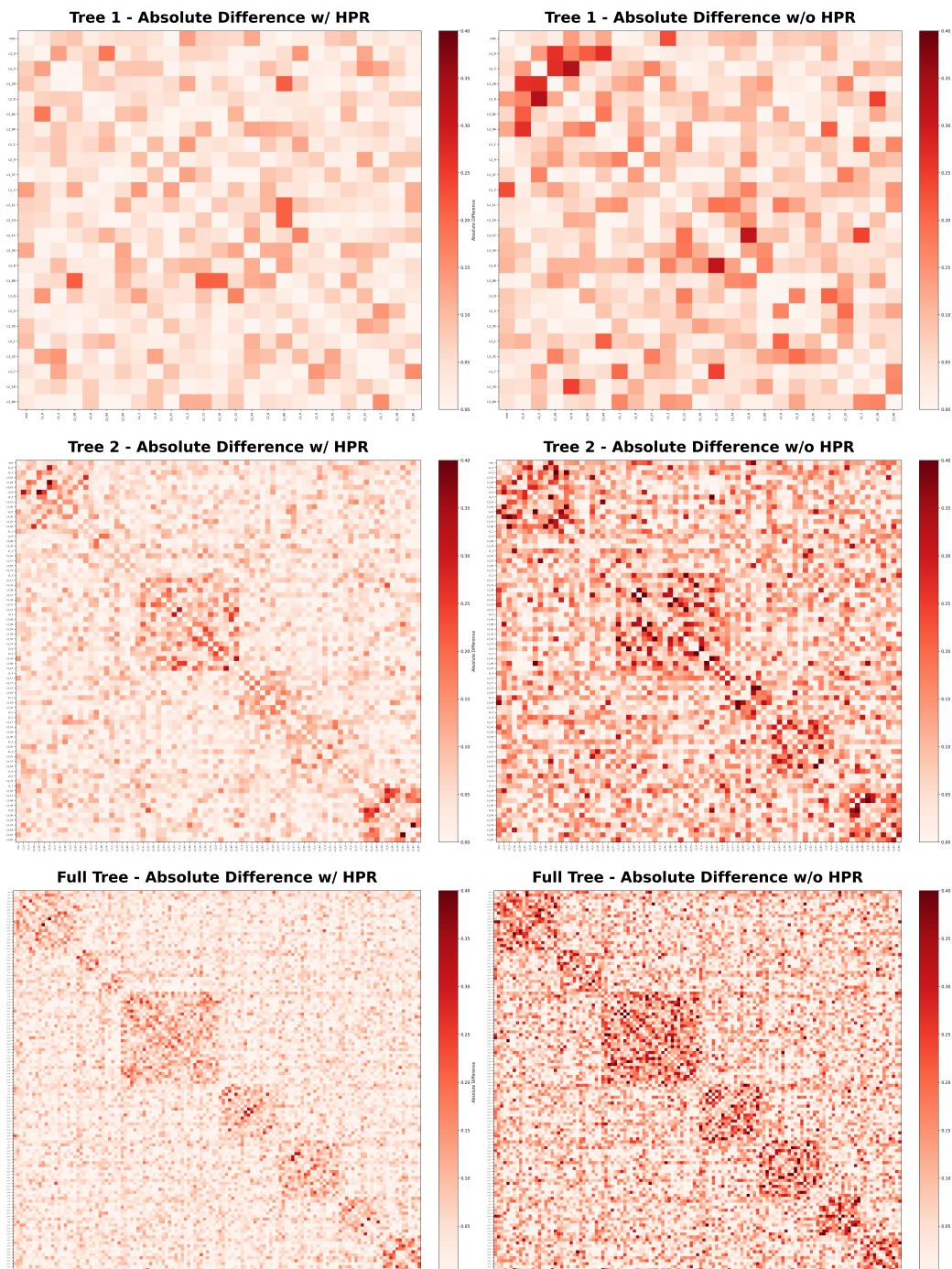

*Figure 13.* Difference matrices between the learned prototype hierarchies and the ground-truth label hierarchy. Each matrix visualizes the deviation from the ground-truth normalized LCA-based distances. The left and right columns correspond to the results obtained with and without HPR. As shown, incorporating HPR notably reduces these discrepancies, providing direct empirical evidence that our proposed module effectively aligns feature representations with the evolving hierarchical tree.

