# OpenReview forum: "Online Continual Learning with Dynamic Label Hierarchies"
_ICML.cc/2026/Conference — ICML 2026 regular_

### Official Review · Reviewer_Cqjk · 2026-03-05

**Soundness:** 3
**Presentation:** 2
**Significance:** 2
**Originality:** 3
**Overall Recommendation:** 4
**Confidence:** 4

**Summary:**

This paper studies online continual learning (OCL) under a hierarchical label space. The authors argue that most existing OCL methods assume a flat label space and therefore cannot properly handle real-world scenarios where classes are organized hierarchically and the taxonomy evolves over time. To address this limitation, the paper introduces a new setting called Dynamic Hierarchical Online Continual Learning (DHOCL), where samples arrive sequentially and labels may appear at different levels of a hierarchy. To tackle this setting, the authors propose HALO (Hierarchical Adaptive Learning with Organized Prototypes). The method combines two key components: (i) Hierarchical Prototype Regularization (HPR), which maintains class-specific prototypes and enforces structural alignment them. (ii) PredLA, a hierarchical prediction aggregation module with two complementary classifiers: a plastic linear classifier trained on evolving features and a stable analytic classifier operating on frozen representations. Experiments are conducted on several image classification datasets (CIFAR-100, FGVC-Aircraft, CUB-200, and iNaturalist) under simulated streaming scenarios with hierarchical labels. The results suggest that the proposed approach improves hierarchical accuracy metrics compared to several baselines.

**Compliance With Llm Reviewing Policy:**

Affirmed.

**Final Justification:**

After reading the rebuttal and the other reviewers’ comments, I found the authors’ clarifications convincing. I revised my score accordingly.

**Key Questions For Authors:**

1. How would the proposed method behave under true structural hierarchy changes, such as merging classes, splitting classes? Have the authors considered evaluating such scenarios?

2. The method assumes access to an external knowledge source to retrieve parent-child relations when new classes appear. Could the authors clarify how this retrieval is implemented in practice?

**Limitations:**

Impact Statement: yes

Limitations: not discussed

**Strengths And Weaknesses:**

Strengths:

1. The paper addresses an interesting and underexplored problem, namely continual learning under evolving hierarchical label spaces. Extending OCL beyond flat label assumptions is a meaningful direction with potential practical relevance.

2. The proposed formulation bridges two previously separate areas: hierarchical classification and online continual learning. This integration could inspire future work at the intersection of these domains.

3. The experimental evaluation covers multiple datasets with different hierarchical structures, which helps demonstrate that the approach is not limited to a single benchmark.

Weaknesses:

1. The paper claims to model real-world taxonomy evolution, but the experimental setup still assumes that the true hierarchy exists beforehand. The dynamics are simulated by splitting classes into streams and randomly selecting label levels. As a result, the hierarchy itself never actually changes structurally during training. This raises concerns about whether the proposed setting truly captures evolving taxonomies. The core claim of the paper is about learning with evolving taxonomies, but in practice the hierarchy does not evolve; it is simply revealed gradually over time, and the granularity level h is already provided during training. In realistic scenarios, taxonomy evolution could involve structural changes such as merging classes or splitting classes, which are not considered here.

2. The framework assumes that the model can query an external knowledge source (e.g., WordNet, expert systems, or LLMs) to retrieve parent-child relations when new classes appear. However, the paper does not explain how this retrieval process works at all. Important aspects remain unclear, such as how relations are retrieved, what kinds of errors occur, and how robust the system is to incorrect relations. Moreover, no experiments evaluate scenarios with incorrect hierarchy edges or hierarchy construction, even though the method claims to operate under evolving taxonomies. This is a significant missing validation.

3. The proposed method introduces many interacting components, including hierarchical prototypes, regularization, temporal saliency constraints, dual classifier heads, prediction aggregation, replay buffer, temperature calibration, and bilevel objective. This level of complexity makes the approach really hard to understand and raises concerns about over-engineering. It is possible that the improvements observed in the experiments stem primarily from the increased model capacity and additional mechanisms rather than from solving the core problem. Moreover, the ablation study only covers PredLA and HPR, but does not isolate the impact of other components in the framework.

4. Many central claims in the paper are not sufficiently supported by empirical evidence. For example, the paper claims that (i) hierarchical prototypes align the feature space with the taxonomy and that (ii) the proposed framework handles evolving taxonomies effectively. However, the paper does not provide any analysis or visualization to support these claims. There is no prototype interpretability analysis, hierarchy embedding visualization, or analysis of how the hierarchy evolves in the learned representation space. The reported metrics primarily demonstrate classification accuracy rather than validating the structural properties that the method claims to enforce.

5. The baselines used in the experimental comparison do not appear to be sufficiently strong for this setting. Most of the compared methods were originally designed for different continual learning settings and it is unclear whether they were carefully adapted to the proposed hierarchical scenario. As a result, their performance appears unusually low. In Table 1, the strongest baseline is ACIL, which does not use a replay buffer or hierarchical information, yet it remains competitive. In fact, HALO shows lower final accuracy on CIFAR-100 and Aircraft despite using a replay buffer of size 1000. With smaller buffers (e.g., 100 as shown in Table 7), the gap could become even larger. Since ACIL is the strongest baseline in the paper, it would also be important to compare with GACL, a follow-up work by the same authors, which is not included in the comparison.

---

> ### Author Rebuttal · Authors · 2026-03-30
>
> **W1&Q1: The hierarchy merging/splitting classes is not considered.**
> (a) Structural evolution occurs: At time $t$, the model observes only a partial subtree. We do not assume tree depth is known beforehand, but require each class's hierarchy level upon arrival (necessary since classification losses and metrics must occur at the same hierarchical level).
>
> (b) Merging/splitting: HALO can handle coexisting hierarchical labels (dog, husky, labrador at different levels). However, we cannot handle label replacement where "dog" disappears and splits into {husky, labrador}—the model must forget dog while simultaneously learning its splits. We view our work as a first step. Label replacement is a substantially complex and interesting problem for future work.
>
> **W2&Q2: External knowledge retrieval and Incorrect relations.**
> **A:** We pre-construct domain-specific hierarchies offline (sources: FGVC/CUB/iNaturalist use native taxonomies, ImageNet from WordNet, CIFAR-100 from GPT-4-refined prior work). During DHOCL, the model does not access the full tree upfront, when a new class $c_j$ arrives, we query $R(c_j) = K(c_j, V_{t-1})$ to retrieve only its parent-child relations with previously seen classes, then incrementally update the hierarchy. This simulates real-world scenarios where taxonomic knowledge becomes available progressively (see Appendix C.4 for delayed-update analysis).
>
> Robustness to hierarchy noise: To directly address concerns about noisy or incomplete hierarchies, we evaluated HALO under corrupted iNaturalist hierarchies with edge deletions (vacant) and random perturbations (wrong):
> | Noise connection | Vacant connection | FFAcc | AAUC | FAUC | MS |
> |-|-|-|-|-|-|
> | 0\% | 0\% | 40.42 | 78.06 | 41.02 | 1.56 |
> | 0\% | 20\% | 40.17 | 76.40 | 41.31 | 1.59 |
> | 0\% | 40\% | 40.23 | 75.88 | 40.76 | 1.61 |
> | 20\% | 0\% | 40.08 | 75.13 | 40.08 | 1.64 |
> | 40\% | 0\% | 40.03 | 73.05 | 39.74 | 1.71 |
>
> (1) HALO degrades gracefully: even with 40% vacant edges, metrics drop <3%. (2) Wrong edges are more disruptive than missing edges (40% noise: −5.01 AAUC vs. 40% vacant: −2.18 AAUC).
>
> **W3: More detailed ablations.**
> **A:** We acknowledge that HALO contains multiple components and understand the over-engineering concern. As a pioneering work introducing the DHOCL setting, we emphasize that the challenges extend far beyond OCL and hierarchical classification. There is no silver bullet for these challenges. Each component addresses specific problems that cannot be solved by simply increasing model capacity. We provide a clear mapping and more detailed ablation study to show each component's purpose and benefit (Due to space limit, please refer to our response to **Reviewer AgvN's W1**).
>
> **W4: Claims about hierarchical alignment lack evidence.**
> **A:** We directly visualize how hierarchical prototypes structurally align with the taxonomy by visualizing learned prototype distances during DHOCL training on CIFAR-100 in Fig.3&4 from link (https://anonymous.4open.science/r/supplementary-EF93/main.pdf). We compute the normalized pairwise distances between class prototypes in HPR's embedding space as $d_{ij} = 1 - \max_{p\in P_i, q \in P_j} \cos(p,q)$ ($ P_i,  P_j$ are prototypes banks for class $i$ and $j$), where the similarity term matches our optimization objective in Eq. 5. We compare these learned distances against ground-truth (ideal) taxonomic distances (measured by normalized LCA depth from the given ground-truth class hierarchy tree). The results show, as HPR is applied, the learned prototype distance matrices become increasingly aligned with the ground-truth hierarchical structure. We genuinely appreciate this valuable critique and hope these analyses address your concerns.
>
> **W5: ACIL's strong performance and GACL.**
> **A:**  ACIL does show strong FFAcc, particularly on datasets where frozen features provide sufficient discriminability which also motivates us to incorporate ACIL heads in PredLA. But its analytic design also limits its extensibility to additional training signals (e.g., hierarchical supervision or replay data) to adapt to more complex scenarios. Meanwhile, **FFAcc alone does not fully characterize hierarchical performance**. As shown in Sec. 3 and Fig. 4, ACIL suffers from coarse-grained degradation and abrupt drops on certain classes despite strong fine-grained retention. We add GACIL comparisons below.
>
> | Dataset | Method | FFAcc | AAUC | FAUC | MS |
> |-|-|-|-|-|-|
> | CIFAR100 | ACIL | 35.3 | 48.7 | 38.8 | 1.36 |
> || GACIL | 36.1 | 51.5 | 39.4 | 1.37 |
> || HALO  | 31.7 | 59.0 | 43.9 |1.12|
> | iNaturalist | ACIL | 26.8 | 67.2 | 34.6 | 2.24 |
> || GACIL | 30.9 | 69.0 | 34.1 | 2.20 |
> || HALO  | 40.1 | 78.3 | 40.8 |1.55|
>
> GACIL improves over ACIL but remains suboptimal on trajectory-aware metrics. On complex iNaturalist taxonomy, HALO outperforms across all metrics validating its superiority. According to your comments, we will add detailed discussion of the ACIL family in the revision.

---

> > ### Author Rebuttal · Reviewer_Cqjk · 2026-04-02
> >
> > Thank you for the rebuttal and the additional clarification. I appreciate the authors’ response and will adjust my score accordingly.

---

> > > ### Author Response · Authors · 2026-04-06
> > >
> > > Thank you very much for your positive evaluation and for your willingness to adjust the score. We are pleased that the additional visualization provided in the rebuttal were helpful. We will faithfully include these points in the revised version to further improve the paper.

---

### Official Review · Reviewer_5kKZ · 2026-03-10

**Soundness:** 3
**Presentation:** 3
**Significance:** 2
**Originality:** 2
**Overall Recommendation:** 4
**Confidence:** 3

**Summary:**

This paper studies online continual learning under dynamically evolving label hierarchies and introduces a new problem setting called Dynamic Hierarchical Online Continual Learning (DHOCL). In this setting, data arrives as a stream and labels may appear at different levels of a hierarchy whose structure can evolve over time. To address the resulting challenges, the authors propose HALO, a framework that combines hierarchical prototype regularization to align representations with the hierarchical structure and a prediction aggregation mechanism that integrates complementary classifiers to balance plasticity and stability. Experiments on several benchmark datasets demonstrate improvements over existing continual learning baselines across multiple hierarchical evaluation metrics.

**Compliance With Llm Reviewing Policy:**

Affirmed.

**Final Justification:**

The authors' rebuttal has properly addressed my concerns.

**Key Questions For Authors:**

See the above Weaknesses.

**Limitations:**

yes

**Strengths And Weaknesses:**

## Strengths

1. The paper introduces a more general continual learning setting, Dynamic Hierarchical Online Continual Learning (DHOCL), which allows labels at different granularity levels to appear in arbitrary order as the hierarchy evolves.

2. The proposed HALO framework is conceptually clear and modular, combining hierarchical prototype regularization with a complementary prediction aggregation mechanism.

3. The experimental evaluation is fairly comprehensive, covering multiple datasets and hierarchical metrics, along with ablation studies to analyze the contributions of different components.

## Weaknesses
1. The proposed approach mainly combines several existing ideas, such as prototype-based representation learning, hierarchical regularization, analytic classifiers, and prediction ensembling. While the integration is reasonable, the method itself does not introduce fundamentally new learning principles, which somewhat limits the novelty of the contribution.

2. The dynamic hierarchy is constructed using external resources such as knowledge bases or expert systems rather than being learned directly from data. As a result, the model does not truly infer hierarchical relations on its own, which weakens the claim of handling dynamically evolving taxonomies.

3. The paper assumes that each sample is annotated at only a single hierarchical level, while labels at other levels are missing. However, many real-world hierarchical datasets already provide full taxonomy annotations. The paper would benefit from stronger justification or real-world examples where this supervision scenario naturally arises.

4. Although the paper compares with several classical continual learning methods, the evaluation lacks some more recent or stronger baselines. Including additional state-of-the-art continual learning or hierarchical learning approaches would help better demonstrate the effectiveness of the proposed method.

---

> ### Author Rebuttal · Authors · 2026-03-30
>
> **(W1) Method mainly combines existing idea**
> **A:** Thank you for feedback but we respectfully argue that our contribution lies in systematically addressing a novel problem rather than proposing entirely new learning primitives. As the first work to study **Online Continual Learning with Dynamic Label Hierarchies**, we began by analyzing the unique challenges along with evolving taxonomies (Sec.3): incompatible classifier heads, heterogeneous learning dynamics across hierarchy levels and representation-taxonomy misalignment. Our solution components are designed to directly address these challenges (see Component Mapping Table in response to Reviewer AgvN's W1).
>
> While individual techniques leverage prior ideas (prototypes, analytic classifiers), their integration and adaptation are non-trivial and new: 1. **Dual-head architecture:** The aggregation balances fine-grained discrimination and heterogeneous learning and forgetting dynamics across hierarchies. 2. **HPR module:** Our regularization (Eq.5) explicitly enforces multi-level taxonomic structure in the feature space.
>
>
> **(W2) Dynamic hierarchy relies on external resources**
> **A:** We acknowledge that our framework relies on externally provided hierarchies rather than inferring them from data.
>
> As a pioneering work in this setting, we focus on the foundational challenge of maintaining and aligning representations with evolving taxonomies, assuming hierarchical knowledge is available (e.g., from domain experts, knowledge bases, or crowdsourcing). We agree that **autonomous hierarchy discovery** would be a valuable extension, but it introduces substantial additional complexity—inferring correct multi-level taxonomic structures from limited streaming data without supervision is an open research problem in itself.
>
> Future direction: We fully recognize the importance and value of your feedback. Enabling models to infer or refine hierarchies with minimal external knowledge, perhaps through self-supervised clustering, uncertainty-based parent assignment, or hybrid human-in-the-loop approaches would significantly enhance practical applicability.
>
> **(W3) Justify annotation assumption**
> **A:** Partial hierarchical labeling is actually pervasive:
> 1. **iNaturalist/GBIF:** Users with varying expertise provide labels at different granularities. Analysis of the iNaturalist-2021 dataset reveals substantial annotation heterogeneity, with observations spanning all taxonomic ranks from kingdom to species. Studies also show annotators naturally label at their confidence level. [Deng J, et al. Large-scale object classification using label relation graphs. ECCV 2014] report over 40% of ImageNet hierarchy annotations are incomplete.
> 2. **Medical diagnosis:** Clinical datasets exhibit annotation heterogeneity—specialists provide fine-grained subtypes (*Invasive Ductal Carcinoma*), general practitioners use broader categories (*Breast Cancer*). Pathology studies report 30-50% coarse-only annotations due to inter-observer variability [Campanella G, et al. Clinical-grade computational pathology using weakly supervised deep learning on whole slide images. Nature Medicine 2019].
>
> In fact, full taxonomy supervision represents an idealized case that rarely occurs (prohibitively expensive at scale), and in OCL, new categories' full taxonomic placement is often initially uncertain (e.g., cryptic species pending genetic analysis). DHOCL also naturally extends to partial multi-level annotations and serves as the lower bound to demonstrate robustness under minimal supervision.
>
> **(W4) More state-of-the-art comparisons.**
> **A:** According to your and Reviewer Cqjk's suggestion, we add experiments on GACIL (the followups of ACIL) to validate the performance of HALO compared with more recent continual method OnLora and analytic method GACIL.
> | Dataset | Method | FFAcc | AAUC | FAUC | MS |
> |-|-|-|-|-|-|
> | CIFAR100 | OnLora | 35.3 | 48.7 | 38.8 | 1.36 |
> || GACIL | 36.1 | 51.5 | 39.4 | 1.37 |
> || HALO  | 31.7 | 59.0 | 43.9 |1.12|
> | iNaturalist | ACIL | 26.8 | 67.2 | 34.6 | 2.24 |
> || GACIL | 30.9 | 69.0 | 34.1 | 2.20 |
> || HALO  | 40.1 | 78.3 | 40.8 |1.55|
>
> As shown above, HALO maintains its advantages. Regarding the lack of comparison with more recent hierarchical classification methods, as we mentioned at the beginning of Sec. 3, most recent hierarchical classification works that focus on improving embeddings and classifiers require continuous restructuring of embedding spaces or model components under evolving taxonomies—operations that are difficult to integrate directly into the OCL setting. In fact, our proposed HPR can already be viewed as a form of hierarchy-aware embedding regularization similar to what many hierarchical classification methods employ. We definietely agree that exploring adaptations of recent hierarchical classification methods to DHOCL would be a valuable direction.

---

> > ### Author Rebuttal · Reviewer_5kKZ · 2026-04-07
> >
> > The authors' rebuttal has properly addressed my concerns. I prefer to keep my score.

---

> > > ### Author Response · Authors · 2026-04-07
> > >
> > > We thank the reviewer for the final assessment and for acknowledging the effectiveness of our rebuttal. We are pleased that the clarifications have resolved the initial concerns, and we will ensure these refinements are integrated into the revised manuscript.

---

### Official Review · Reviewer_5bUH · 2026-03-12

**Soundness:** 3
**Presentation:** 3
**Significance:** 3
**Originality:** 3
**Overall Recommendation:** 5
**Confidence:** 4

**Summary:**

This paper introduces Dynamic Hierarchical Online Continual Learning (DHOCL), which extends standard Online Continual Learning (OCL) by allowing label hierarchies to evolve dynamically over time while data instances arrive annotated at arbitrary granularity levels.

Unlike prior hierarchical continual learning settings such as HLE (Lee et al., 2023) and IIRC (Abdelsalam et al., 2021), which assume a rigid coarse to fine curriculum, DHOCL permits labels at any hierarchical depth to appear at any time, and the taxonomy itself grows organically as new classes are encountered. The hierarchy is constructed on the fly by querying an external knowledge source whenever a new class arrives.

**Compliance With Llm Reviewing Policy:**

Affirmed.

**Final Justification:**

In the light of the rebuttal, i am increasing the score, as it addresses all my concerns.

**Key Questions For Authors:**

1. **Hierarchy robustness:** How does HALO perform when the external knowledge source K introduces errors in the retrieved hierarchy (e.g., incorrect parent child assignments, missing links)? A controlled experiment with synthetic hierarchy noise at varying error rates would substantially strengthen the practical claims.

2. **Data duplication fairness:** Can you confirm that all 16 baselines in Table 1 receive the identical duplicated data streams (2x for Aircraft/CUB, 4x for iNaturalist)? If so, have you verified that the relative rankings hold under a strict single pass protocol, even if absolute numbers are lower?

3. **Confidence intervals:** Given that five seeds were run, can you provide standard deviations for all results in Table 1? Several margins appear small, and it would be valuable to know which improvements are statistically significant.

4. **Aggregation weight dynamics:** How do the learned PredLA weights $alpha_l^h$ and $\alpha_a^h$ evolve over training for different hierarchy levels? Visualizing these trajectories would provide insight into whether the method truly learns to shift between plasticity and stability at different levels, as claimed.

5. **Hyperparameter transfer:** Were the default hyperparameter values (gamma=1.0, |P_c^h|=5, m=0.1, K=top 10%) selected on the test set or on a held out validation split? If the former, how sensitive are the results to suboptimal choices, particularly for new domains not included in the evaluation?

6. **Differentiation from prior components:** How does HALO's dual head PredLA design differ from CCL DC (Wang et al., CVPR 2024), which also uses complementary stable/plastic classifier heads for online CL? What specific benefit does the hierarchical prototype structure provide over ICICLE's (Rymarczyk et al., ICCV 2023) prototype based CL approach? Explicit ablations or comparisons isolating these differences would strengthen the novelty claim.

**Limitations:**

The authors briefly acknowledge in the conclusion (Section 6) that "our setting still falls short of full real world complexity (e.g., skewed or corrupted tree structures)." This is an honest treatment.

**Strengths And Weaknesses:**

## Strengths and Weaknesses

### Strengths

**S1 [Originality/Significance].** The DHOCL problem formulation is novel and well-motivated. The paper convincingly argues (Section 2) that real-world taxonomies evolve inductively and that annotation granularity depends on observer expertise, making the rigid coarse-to-fine assumption of HLE/IIRC unrealistic. The three factors cited  , expertise diversity, inductive growth, and structural openness  , are compelling. Figure 1 provides an effective visual comparison of OCL, HLE/IIRC, and DHOCL settings.

**S2 [Soundness].** The diagnostic analysis in Section 3 is thorough and informative. The systematic evaluation of five hierarchical-aware losses (Figure 2) and four anti-forgetting strategies (Figures 3 ,4) across four datasets provides strong empirical evidence for why existing methods fail in DHOCL. The identification of "supervisory jitter"  , abrupt shifts in loss targets caused by taxonomic topology changes  , and the observation of heterogeneous learning/forgetting dynamics across hierarchy levels (particularly the ACIL comparison showing its complementary strengths and weaknesses) are insightful and well-supported.

**S3 [Significance/Soundness].** The experimental evaluation is comprehensive. Table 1 compares against 16 baselines spanning replay-based (RS, CBRS, MIR, CLIB, RM, PLFMS), regularization-based (LwF, EWC++, ICICLE), prototype-based (iCaRL w/ SDC, OnPro), analytic (ACIL), and recent OCL methods (NsCE, CCL-DC, OnLora) across four datasets and five metrics (AAUC, FAUC, MS, FFAcc, FAAcc). HALO achieves the best results across all benchmarks and nearly all metrics. The cross-architecture evaluation (Table 3) covering ResNet-50 and ViT-B with five pretraining strategies (Supervised, CLIP, DINO, MAE) further strengthens the generality claim.

**S4 [Presentation].** The paper is generally well-written and logically structured. The progression from problem definition (Section 2) to diagnostic analysis (Section 3) to method (Section 4) to experiments (Section 5) builds a clear narrative. The evaluation protocol with five complementary metrics (AAUC, FAUC, MS, FFAcc, FAAcc) is well-designed to capture different aspects of hierarchical continual learning performance.

**S5 [Significance].** The ablation study (Table 2) cleanly decomposes the contributions of PredLA and HPR. PredLA alone accounts for the majority of gains (e.g., CUB-200 AAUC from 52.5 to 61.2), while HPR provides consistent but smaller additional improvements (e.g., CUB-200 AAUC from 61.2 to 62.9, iNaturalist FFAcc from 35.2 to 40.1), suggesting both modules contribute meaningfully but that the aggregation mechanism is the primary driver.

### Weaknesses

**W1 [Soundness]. (Major)** The paper assumes access to a perfect external hierarchy oracle K (Section 2, Eq. R(c_j) = K(c_j, V_{t-1})). In practice, WordNet coverage is incomplete for many domains (e.g., fine-grained aircraft variants), and GPT-generated hierarchies can be noisy or inconsistent. The paper acknowledges this only briefly in the conclusion ("our setting still falls short of full real-world complexity, e.g., skewed or corrupted tree structures") but does not evaluate robustness to hierarchy noise or errors in the retrieved relations. This is a significant practical limitation given that DHOCL is motivated precisely by real-world deployment scenarios.

**W2 [Soundness]. (Major)** I am seriously concerned about the data duplication strategy in Appendix B.1. It fundamentally violates the single-pass assumption that defines OCL: the stream is duplicated 2x for FGVC-Aircraft and CUB-200, and 4x for iNaturalist.

While all baselines share the same duplicated stream, this disproportionately favors HALO, whose prototype learning (HPR, Eq. 4) and bilevel aggregation weight optimization (PredLA, Eq. 12) both require sufficient per-class observations to converge, whereas simpler baselines like RS or EWC++ are less sensitive to sample frequency. No ablation under a true single-pass setting is provided, making it impossible to determine whether HALO's gains reflect genuine architectural advantages or a more favorable sample regime for its more complex objectives.


**W3 [Presentation/Soundness].** While Figure 6 shows sensitivity analysis for lambda and delta, and Figure 8 (Appendix B.3) covers the remaining hyperparameters, most are evaluated only on AAUC. The cross-hyperparameter interaction effects are not studied. The stated defaults (gamma=1.0, |P_c^h|=5, m=0.1, K=top-10%) are used uniformly across all datasets, but it is unclear whether these were selected via held-out validation or tuned on the test metrics. The sheer number of hyperparameters raises concerns about the method's practical deployability.


**W4 [Significance].** The throughput analysis (Figure 10, Appendix C.3) reveals that HALO processes only 266.7 samples/second on CIFAR-100, compared to 1241.4 for the Analytic baseline and 637.8 for RS(HAF). This represents a roughly 2.4x slowdown compared to the basic replay baseline and a 4.7x slowdown compared to analytic methods, which is substantial for an online learning setting where efficiency is often critical. This computational overhead, driven primarily by the dual-head architecture and prototype computations, somewhat undermines the practicality claim.


**W5 [Originality/Presentation].** The handling of imbalanced hierarchies (ImageNet-H, Appendix B.1 and Table 8) relies on a "randomized padding strategy" that duplicates ancestor nodes to force a temporarily balanced form. I feel that this is an ad hoc workaround that the authors themselves acknowledge introduces evaluation artifacts ("comparisons across categories at slightly different semantic granularities"). A more principled treatment of imbalanced hierarchies would strengthen the contribution.

---

> ### Author Rebuttal · Authors · 2026-03-30
>
> **W1&Q1: Incorrect relations.**
> **A:** Due to space constraints, please see our response to Reviewer **Cqjk's W2&Q2** for similar questions.
>
> **W2&Q2: Results under strict single-pass protoco.**
> **A:** We confirm that all 16 baselines receive identical duplicated streams (2 for Aircraft/CUB, 4 for iNaturalist), ensuring fair comparison. The duplication was necessary because strict single-pass settings cause severe underfitting across all methods—even ACIL (frozen features) shows significant degradation, indicating the extreme supervision sparsity in DHOCL makes meaningful comparison difficult.
>
> To directly address your concern, we re-ran key experiments under strict single-pass protocol. Results confirm that most relative rankings remain consistent, even though absolute numbers are quite low:
>
> | Dataset | Method | FFAcc | AAUC | FAUC | MS |
> |-|-|-|-|-|-|
> | FGVC | ER | 4.17 | 8.00 | 16.20 | 2.03 |
> | | EWC++ | 4.41 | 10.23 | 18.05 | 2.11 |
> | | ACIL | 17.26 | 14.33 | 22.12 | 1.79 |
> | | CCL-DC | 16.22 | 23.43 | 24.14 | 1.72 |
> | | HALO w/o HPR| 26.47 | 26.86 | **28.10** | 1.51 |
> | | HALO | **26.51** | **27.74** | 28.02 | **1.46** |
> | CUB | ER | 2.77 | 34.67 | 5.70 | 1.80 |
> | | EWC++ | 4.01 | 37.05 | 8.04 | 1.76 |
> | | ACIL | 26.06 | 46.21 | 21.00 | 1.38 |
> | | CCL-DC | 23.41 | 48.13| **36.52** | 1.12 |
> | | HALO w/o HPR| **28.84** | 51.13 | 36.04 | 1.07 |
> | | HALO | 28.67 | **52.70** | 36.30 | **0.96** |
> | iNaturalist | ER | 2.10 | 50.57 | 4.75 | 3.31 |
> | | EWC++ | 4.57| 51.02 | 8.32 | 3.17|
> | | ACIL | 12.98 | 56.29 | 17.95 | 3.03 |
> | | CCL-DC | 16.12 | 65.01| 28.52 | 2.47 |
> | | HALO w/o HPR| 17.31 | 66.32 | 26.01 | 2.34 |
> | | HALO | **18.95** | **68.42** | **28.75** | **2.16** |
>
> Regarding sensitivity to sample frequency: PredLA optimizes only single scalar weights (convergence is not an issue), demonstrating genuine architectural advantages rather than data-regime dependency. While HPR may indeed benefit from more samples for prototype learning. However, its gains on trajectory-aware metrics (AAUC/MS) persist. These results verify HALO's superiority stems from addressing hierarchical forgetting dynamics, not merely favorable sample regimes.
>
> **W3&Q5: Hyperparameter selection process.**
> **A:** As detailed in our supplementary code, we use a held-out validation set (20% of training data) for hyperparameter tuning on all datasets except CUB (For CUB we simply use default setting). Meanwhile, most hyperparameters follow established defaults from prior work (e.g., $|P_c^h|=5$, $K=10\%$ from ICICLE). Figure 8 shows HALO maintains stable performance across wide ranges. Critically, **we performed no per-dataset tuning or cross-hyperparameter optimization**. The same defaults work uniformly across all datasets with diverse characteristics. For new domains, we recommend: (1) use our defaults, (2) validate only $\lambda$ and $\delta$ (most task-dependent) on a small held-out split if needed.
>
> **W4: Lower throughput.**
> **A:** Due to space limit, we kindly refer you to our response to similar questions raised by **Reviewer AgvN (W2)**.
>
> **W5: Ad-hoc strategy for ImageNet-H.**
> **A:** We acknowledge that the randomized padding strategy is an ad hoc solution. The most direct approach is to assign new class to a specific hierarchy level upon arrival but it can be very hard without external knowledge.
>
> **Future direction:** A more principled solution could leverage hierarchical clustering on learned feature embeddings to automatically assign levels. However, this introduces additional complexity that warrants dedicated investigation beyond the current scope. We will clarify this in the revision and explore feature-based assignment in future work.
>
> **Q3: Confidence Interval**
> **A:** We provide standard deviations for the majority of methods in  Tab.1 (https://anonymous.4open.science/r/supplementary-EF93/main.pdf) to assess statistical significance. Due to cluster transitions, we were unable to preserve complete raw results for all baselines across all five seeds. Nevertheless, we have provided standard deviations for as many methods as possible to enable transparency.
>
> **Q4: Visualization of PredLA weight evolution**
> **A:** In Fig.1 of (https://anonymous.4open.science/r/supplementary-EF93/main.pdf), we provide the visualization of PredLA weight evolution. It shows that as training progresses, the weights shift toward a level-dependent balance: higher levels (L1) exhibit increased plasticity ($\alpha_l^h$) for flexible adaptation, while lower levels (L7) maintain greater stability (larger $\alpha_a^h$) to preserve fine-grained knowledge. This adaptive weighting mechanism demonstrates the model's ability to balance plasticity and stability across the hierarchy.
>
> **Q6: Explicit comparison with CCL-DC and ICICLE to clarify novelty**
> **A:** We address this concern with detailed ablation analysis in our response to **Reviewer AgvN's W1**.

---

> > ### Author Rebuttal · Reviewer_5bUH · 2026-04-02
> >
> > Thank you for detailed reply. It addresses all my concerns. I would like to increase my score to 5,

---

> > > ### Author Response · Authors · 2026-04-06
> > >
> > > We would like to express our sincere gratitude for your positive feedback and for increasing the score. We are glad that our detailed response and the additional experiments addressed all your concerns.

---

### Official Review · Reviewer_AgvN · 2026-03-16

**Soundness:** 3
**Presentation:** 3
**Significance:** 3
**Originality:** 3
**Overall Recommendation:** 5
**Confidence:** 4

**Summary:**

This paper studies the online continual learning (OCL) problem, where models learn from non-stationary data streams. It proposes that existing methods fail to consider the hierarchical label space, which is more common in the real-world scenarios. To overcome this limitation, the authors propose a new problem setting called online continueal learning from dynamic hierarchies (DHOCL). The authors discover that classic methods face challenges with partial supervision and may fail to retent knowledge under this setting. To address these problems, the authors propose a new method called HALO, which learns hierarchical prototypes to ensure hierarchical consistency and to store hierarchical knowledges. A hierarchical adaptive ensemble module PredLA is further proposed. The proposed method is validated on multiple datasets and perform consistently better than other methods.

**Compliance With Llm Reviewing Policy:**

Affirmed.

**Final Justification:**

The rebuttal has properly addressed my concerns. I would like to keep my positive score.

**Key Questions For Authors:**

See weaknesses.

**Limitations:**

The authors have discussed the limitations in some section. It is suggested that the authors can discuss these limitations in a specific section.

**Strengths And Weaknesses:**

The paper have multiple strength:
- This paper proposes a novel and realistic problem setting, which is more important but neglected by previous works.
- The proposed method is elegant. The core idea is to utilize prototypes to tackle the hierarchical issue, which can solve both the classification and the regularization problems.
- The authors have conducted thorough experiments, including ablation studies and sensitivity analyses to demonstrate the effectiveness.
- The paper is well-written and easy to follow. The proposed method is formulated well, and the training procedure is given for easy understanding.

Besides, there are also some weaknesses:
- The proposed method introduce multiple components, including an add-on adapter, class-specific prototype banks, temperature scaling, aggregation weights. It is better to highlight the connection and the necessity of each component.
- Adding discussion of computational overhead compared to baseline methods can further improve the quality of the paper.
- The proposed method performs sub-optimal on some metrics such as FFAcc. It is better to discuss the potential reason, or clarify the importance of these different metrics.

---

> ### Author Rebuttal · Authors · 2026-03-30
>
> **W1: The proposed method introduce multiple components, it is better to highlight the connection and the necessity of each component.**
>
> **A:** Thank you for this important question. We acknowledge the complexity and provide a direct component-to-challenge mapping with comprehensive ablations.
>
> Component mapping and motivation: Each component addresses specific challenges identified in Sec. 3:
>
> | Component | Challenge Tackled | Design Rationale |
> |-|-|-|
> | **PredLA** | Prediction aggregation | Coordinates learning/forgetting across granularities |
> | ↳ Bilevel temperature calibration (Eq. 12) | Overconfident linear head dominance | Prevents ACIL head from overwhelming HALO predictions |
> | **HPR** | Temporal saliency consistency | Naive fine-grained forgetting prevention (hierarchical extension of ICICLE with add-on adapters + class-specific prototype banks) |
> | ↳ Hierarchical structural alignment | Evolving taxonomy matching | Aligns growing label trees in prototype embedding space |
>
> Extended ablation study: We perform detailed ablations on iNaturalist (more complex hierarchy: Height=7) to isolate each component's contribution:
>
> | Configuration | AAUC | FAUC | MS | FFAcc |
> |--------------|------|------|-----|-------|
> | PredLA w/o bilevel calibration | 72.1 | 26.4 | 1.96 | 16.4 |
> | PredLA (with calibration) | 76.5 | 39.0 | 1.63 | 35.0 |
> | + Temporal saliency | 77.0 | 41.3 | 1.60 | 40.3 |
> | HALO (+ Hierarchical alignment) | 78.5 | 41.2 | 1.54 | 39.8 |
>
> Key observations:
> - Without bilevel calibration, PredLA degenerates to linear classifier dominance (16.4% FFAcc), confirming that Eq. 12 is essential for organic fusion of dual predictions and coordinating cross-granularity learning.
> - Temporal saliency significantly improves forgetting resistance, especially on FFAcc (35.0→40.3%), demonstrating fine-grained retention benefits.
> - Hierarchical alignment provides targeted improvements on hierarchy-aware metrics (AAUC: 77.0→78.5, MS: 1.60→1.54), validating its role in hierarchical consistency.
>
> Comparison with related work:
> - vs. CCL DC: While CCL DC also carefully designs their classification heads, it targets the plasticity-stability dilemma in flat CL. The analytic head we use in HALO presents unique advantages in knowledge retention, as Reviewer Cqjk mentioned. HALO's PredLA combines these merits and successfully addresses hierarchical granularity conflicts.
> - vs. ICICLE: ICICLE uses flat prototypes for single-level CL. HPR extends this with hierarchical alignment term that explicitly model parent-child relationships and temporal evolution across taxonomy levels which is absent in ICICLE.
>
> We will add these comparisons and clarify component necessity in revision.
>
> **W2: Adding discussion of computational overhead compared with baselines**
> **A:** We first acknowledge the computational overhead and clarify the accuracy-efficiency trade-off. HALO achieves 266.7 samples/s on CIFAR-100, representing a 2.4× slowdown vs. RS (637.8 samples/s) and 4.7× vs. analytic methods (1241.4 samples/s). Throughput measurements were obtained using RTX 4090 GPUs and Intel i9-13900KF CPU. This overhead primarily stems from the hierarchical prototype regularization (HPR) module. As shown in Appendix C.3, HALO without HPR achieves 621.4 samples/s—comparable to other replay baselines—while still outperforming prior methods on hierarchy-aware metrics.
>
> Practical trade-off: The HPR overhead (2.3× slowdown) buys substantial averaged accuracy gains: +5.35% AAUC and -0.29 MS improvement over the second best method. For applications where hierarchical consistency is critical (e.g., biological taxonomy, medical diagnosis), this trade-off is justified. For throughput-sensitive scenarios, practitioners can disable HPR while retaining HALO's core dual-head architecture.
>
> We will expand this discussion in the main paper to help practitioners make informed decisions based on their constraints.
>
>
> **W3: The proposed method performs sub-optimal on FFAcc.**
>    Thank you for the observation. Methods like ACIL that freeze the feature extractor naturally avoid feature-level forgetting, giving them an advantage on FFAcc—a metric that captures only the final snapshot at the finest granularity, where pretrained features often suffice for simple classification tasks. However, ACIL performs less satisfactorily on AAUC/MS, which measure the full learning trajectory across all hierarchy levels. This aligns with our Section 3 observation that learning and forgetting rates vary across granularities, motivating our HPR to enforce detailed feature-topology correspondence. As Table 2 shows, HALO's holistic optimization on full trajectory across all levels and timesteps yields much larger gains on AAUC/FAUC/MS.

---

> > ### Author Rebuttal · Reviewer_AgvN · 2026-04-02
> >
> > The authors' rebuttal has properly addressed my concerns. I prefer to keep my score.

---

> > > ### Author Response · Authors · 2026-04-06
> > >
> > > We thank the reviewer for the final assessment and for acknowledging the effectiveness of our rebuttal. We are pleased that the clarifications have resolved the initial concerns, and we will ensure these refinements are integrated into the final manuscript.

---

### Decision · Program_Chairs · 2026-04-30

**Decision:**

Accept (regular)

**Comment:**

This paper proposes DHOCL, a novel and practical setting for online continual learning with dynamically evolving, mixed-granularity label hierarchies, and introduces HALO to address its unique challenges of hierarchical consistency and granularity-dependent forgetting.

HALO's key strengths are its well-motivated design, combining hierarchical prototype regularization for structural alignment and a dual-head prediction aggregation for balanced learning. The evaluation is thorough, showing consistent gains over many baselines. Initial reviewer concerns about complexity and assumptions were convincingly resolved in a strong rebuttal with new experiments and analysis.

Given its significant novelty, technical soundness, and potential impact, I recommend accept.